# Intrinsic Disorder of the BAF Complex: Roles in Chromatin Remodeling and Disease Development

**DOI:** 10.3390/ijms20215260

**Published:** 2019-10-23

**Authors:** Nashwa El Hadidy, Vladimir N. Uversky

**Affiliations:** 1Department of Molecular Medicine, Morsani College of Medicine, University of South Florida, 12901 Bruce B. Downs Blvd. MDC07, Tampa, FL 33612, USA; nashwa@health.usf.edu; 2Laboratory of New Methods in Biology, Institute for Biological Instrumentation of the Russian Academy of Sciences, Federal Research Center “Pushchino Scientific Center for Biological Research of the Russian Academy of Sciences”, Pushchino, 142290 Moscow Region, Russia

**Keywords:** ATP-dependent nucleosome remodeler, BAF complex, chromatin remodeling, intrinsically disorder protein, intrinsically disordered protein region

## Abstract

The two-meter-long DNA is compressed into chromatin in the nucleus of every cell, which serves as a significant barrier to transcription. Therefore, for processes such as replication and transcription to occur, the highly compacted chromatin must be relaxed, and the processes required for chromatin reorganization for the aim of replication or transcription are controlled by ATP-dependent nucleosome remodelers. One of the most highly studied remodelers of this kind is the BRG1- or BRM-associated factor complex (BAF complex, also known as SWItch/sucrose non-fermentable (SWI/SNF) complex), which is crucial for the regulation of gene expression and differentiation in eukaryotes. Chromatin remodeling complex BAF is characterized by a highly polymorphic structure, containing from four to 17 subunits encoded by 29 genes. The aim of this paper is to provide an overview of the role of BAF complex in chromatin remodeling and also to use literature mining and a set of computational and bioinformatics tools to analyze structural properties, intrinsic disorder predisposition, and functionalities of its subunits, along with the description of the relations of different BAF complex subunits to the pathogenesis of various human diseases.

## 1. Introduction

### 1.1. Role of BAF Complex in Chromatin Remodeling

The two-meter-long DNA is compressed into chromatin in the nucleus of every cell, which serves as a significant barrier to transcription [1]. Therefore, for processes such as replication and transcription to occur, the highly compacted chromatin must be relaxed. There are two major mechanisms of chromatin reorganization for the aim of replication or transcription that alter chromatin structure through disruption of histone-DNA contacts, (i) various covalent modifications of histone tails, such as acetylation, methylation, phosphorylation, ubiquitination, sumoylation, and ADP ribosylation [2,3] and utilization of ATP-dependent chromatin remodeling complexes that use the energy of ATP hydrolysis to disturb the DNA-histone interactions to increase the accessibility of the nucleosomal DNA [4]. Among such ATP-dependent nucleosome remodelers, which are considered important players of transcriptional regulation of many eukaryotic genes, are polymorphic BRG-/BRM-associated factor (BAF) and polybromo-associated BAF (PBAF) complexes. These complexes are the members of the SWItch/sucrose non-fermentable (SWI/SNF) family of the ATP-dependent chromatin-remodeling complexes that are essential for mammalian transcription and development [5]. Although the BAF complex plays an important role in the process of chromatin reorganization for the aim of replication or transcription [1], until quite recently, the detailed mechanism of the action of the complex was not fully understood. However, recent studies suggested that the ATP-dependent chromatin-remodeling complexes might be performing their biological activities through nucleosome exchange [6]. Studies have shown that BAF complexes generally do not bind at promoters, but occupy nucleosome assemblies instead, which is further explained by the “loop recapture” mechanism [6]. In this loop recapture model, an initial DNA loop or bulge is formed by the effect of a translocase activity of the SWI/SNF complex [6,7], where the ATPase binds to a certain region on the nucleosome and uses its translocase activity to move the DNA from one entry or exit to another in a directional wave. It was suggested that this step usually happens in a region of DNA weakness, and that torsional strain is involved in the proliferation of the loop. However, this model is restricted by some limitations as to whether torsional strain is even necessary due to a lack of the effects of nicks and gaps on the process and the low bending flexibility of DNA, thus making the concept of bulge formation questionable [6,7]. Nevertheless, the binding of the complex to the nucleosome causes severe rearrangement of the DNA regarding the histone octamer [6]. In addition, the BAF complex plays a role in the steadfastness of higher order chromatin, which occurs due to the ability of this complex to bind to actin filaments through the initial interaction with the phosphatidylinositol 4,5-bisphosphate (PIP2) micelles [8].

BAF complex also plays an essential role in regulation of gene transcription through its instructive role in gene expression in certain types of cells via its interaction with certain transcription factors [9]. An illustrative example is given by the differentiation of multipotent neuroepithelial cells into the highly specialized neuronal and glial cell types that are present within the adult central nervous system [10]. Furthermore, the BAF complex plays a role in controlling the pluripotency of embryonic cells, where it is attached to almost all of the promoters of almost all the genes involved in the pluripotency, and helps in refining the transcription of the genes involved in pluripotency and self-renewal [9]. BAF is also important for the regulation of the immune response against viral infection through elevating the levels of expression of *INF* genes, thereby leading to the initiation of antiviral activities [10].

In addition to the role of the BAF complex in chromatin remodeling, many subunits of BAF or PBAF, including BRG1, BAF45A, BAF53A, BAF45A, BAF180, and BAF250A, are known to be involved in hematopoiesis. Furthermore, BAF200, which is one of the PBAF subunits, plays a major role in heart morphogenesis and coronary heart angiogenesis [11].

### 1.2. Structure of BAF Complex

The ATP-dependent chromatin remodeling complexes are large multi-component proteinaceous machines that are composed of two to 17 subunits [12] and are highly conserved within eukaryotes. They are characterized by the presence of ATPase subunit that belongs to superfamily II of the helicase-related proteins [12,13]. Based on the unique differences that are present within or adjacent to the ATPase domain, this class can be further divided into four main families: SWI/SNF (the most complex member of the family containing of up to 17 subunits encoded by 29 genes, including several products of the *SWI* and *SNF* genes (*SWI1*, *SWI2/SNF2*, *SWI3*, *SWI5*, and *SWI6*) and other proteins, see below); imitation SWItch (ISWI, containing one ATPase subunit (SMARCA5 or SMARCA5) and one or more accessory subunits); nucleosome remodeling deacetylase (NURD/Mi-2/CHD, including proteins with both ATP-dependent chromatin remodeling and histone deacetylase activities); and INO80 (another large chromatin remodeling complex composed of >15 subunits) [12]. Figure 1 compares the domain organization of the ATPase subunits found in ATP-dependent chromatin remodeling complexes and shows that these ATPase always include a DExx and a HELICc domain, separated by a linker [12,13]. The SWI/SNF family contains a HSA domain, involved in actin binding, and a bromodomain important for the binding of acetylated lysines [12]. A characteristic feature of the ATPases of the ISWI family is the presence of the SANT and SLIDE domains, important for histone binding, whereas histone binding of the CHD/NURD/Mi-2 family is achieved via a tandem of chromo domains located at the N-terminal region of these ATPases. Finally, similar to the SWI/SNF family, ATPases of the INO80 family contain a HSA domain but do not possess a bromodomain and have a longer linker between the DExx and the HELICc domains [12].

One should also keep in mind that the ATP-dependent chromatin remodeling complexes differ from each other not only by the domain organization of their catalytic subunits, but also by their subunit compositions. This is illustrated by Figure 2, showing schematic representations of some of these important complexes.

In mammals, multiple forms of SWI/SNF complexes are present and are composed of 9–12 core proteins that are referred to as BRG1 associated factors (BAF), ranging from 47 to 250 KD [15]. In addition, SWI/SNF complexes may contain brahma homolog (hBRM) [16]. Both BRG1 and hBRM are orthologs of the yeast SWI2/SNF2-like ATPase, and the corresponding ATP-dependent chromatin remodeling complexes in mammals are called either BRG1- or hBRM-associated factors (BAFs) [16]. Based on their subunit organization, SWI/SNF complexes in mammals are further subdivided into two main complexes known as BAF (BRG-/BRM-associated factor, also known as canonical BAF, cBAF) and PBAF (polybromo-associated BAF) that differ from each other based on their subunit compositions, with BAF250A, BAF250B, or BAF180 found in BAF/cBAF or PBAF, respectively [17]. Recently, another type of SWI/SNF complex was described, namely the non-canonical BAF (ncBAF) complex, which lacks the dedicated cBAF subunits and some PBAF-specific subunits, but includes distinctive subunits GLTSCR1/1L [18]. The canonical BAF complex contains a catalytic subunit (either SMARCA4/BRG1/BAF190A or SMARCA2/BRM/BAF190B) and at least SMARCE1/BAF57, ACTL6A/BAF53, SMARCC1/BAF155, SMARCC2/BAF170, and SMARCB1/SNF5/BAF47. Furthermore, even within the BAF complex itself, there is a noticeable polymorphism serving as an important indication of the effect of combinatorial assembly in functional specificity of this regulatory machine. Illustrative examples of distinct subunit compositions taking place at different time points during development and in different tissues are given by embryonic stem (ES) cell-specific BAF (esBAF) needed for the regulation of the ES cell (ESC) transcriptome, the neural progenitor BAF (npBAF) complex that acts at the differentiation from ESCs to neural stem cells (NSCs), and the neuronal BAF (nBAF) complex that controls various aspects of neural development and plasticity [19]. It was pointed out that distinct switching of the BAF complex subunits leading to the transitioning from esBAF to npBAF and to nBAF represents an adaptation of this SWI/SNF complex to the changing necessities of more differentiated cell types [19].

Many different types of BAF complexes were identified, with several core subunits, such as BRG1 (or hBRM), BAF170, BAF155, and BAF47/INI1, as well as BAF60, BAF57, BAF53, and β-actin being consistently found in most purified SWI/SNF complexes. These core subunits of the BAF complexes are responsible for essential chromatin remodeling activity [20]. On the other hand, there are many unique non-core subunits in the BAF complexes with the corresponding orthologs being not found in yeast. The roles of the non-core subunits in targeting or signaling are still unclear [8]. Curiously, it was pointed out that actin acts as one of the integral component of the mammalian BAF and PBAF complexes, being tightly bound to the BAF complex [16,21] via direct interaction with at least two different domains in the C-terminus of the BRG1 protein [8]. In fact, actin is so tightly bound to BAF that this complex can only be eliminated by denaturing conditions [8]. Furthermore, one of the BAF complex subunits, BAF53 shares 38% sequence identity with actin [8,21].

### 1.3. Deregulation of BAF Complex Associated with Diseases

SWI/SNF complex has been linked to several human diseases, mostly different types of cancer. In fact, mutations, translocations, and deletions occurring within some subunits of SWI/SNF complex were found in ~20% of human tumors, making the complex one of the most commonly affected hallmarks in cancer [17,22]. Furthermore, SWI/SNF complexes have been shown to function as both tumor suppressors and oncogenes [17]. Mutations in the SWI/SNF complex are usually associated with loss of protein function, with SMARCB1/BAF47, SMARCA4/BRG1, and ARID1A/BAF250A subunits of the complex being characterized by very high mutation frequencies and a strong relation to disease development [17]. The roles of mutations in the individual BAF-complex subunits in the development of various human diseases will be discussed later.

An illustrative example of a linkage of BAF complex to disease is given by Ewing sarcoma, which is one of the most common bone tumors in children that, along with osteosarcoma, accounts for 6% of childhood malignancies [23]. The femur is the site of this tumor that frequently arises in the midshaft [24]. Other tumors with similar histology, such as peripheral primitive neuroectodermal tumor, Askin tumor, and neuroepithelioma, together with Ewing sarcoma, are collectively known as the Ewing sarcoma family of tumors (ESFT) [24]. The most frequent gene translocation associated with ESFT is t(11;22) (q24;q12) translocation, where fusion of *EWS* gene on 22q12 with *FLI1* gene on 11q24 will produce EWS-FLI1 hybrid protein, and this fusion transcript will lead to increased transcription capacity compared with FLI1 alone [24]. Several factors affect activity of EWS-FLI1 in Ewing sarcoma and in p53 and INK4A pathways, which are crucial in promoting cell cycle. Mutations within these pathways lead to tumor progression [25,26] and unstable EWS-FLI1 expression. Since under the normal conditions, the p53 initiates cell cycle arrest and apoptosis, therefore, a loss of p53 is associated with the enhancement of EWS-FLI1 expression [25,27]. Furthermore, it was shown that in Ewing sarcoma, hypoxia can lead to apoptosis resistance, chemotherapy resistance, and to the establishment of alternative circulatory system [25,28,29]. The resistance to apoptosis is maintained by activation of hypoxia inducible factors (HIF), which are HIF1, HIF2, and HIF3 that regulate transcription in hypoxic environments [30]. Another pathway that maintains the oncogenic activity of the EWS-FLI1 fusion protein is IGF1/IGF1-R and it was proven that IGF1/IGF1-R inhibition increased chemo sensitivity [25,31,32]. When it comes to the role of BAF complex in development of Ewing sarcoma, it was found that the BAF complex binds to EWSR1 protein [33,34], which is one of the RNA binding proteins containing prion like domains [35], and it is the EWS-FLI1 fusion that maintains this binding capability [33]. The interaction between the BAF complex and EWS-FLI1 fusion is weak and transient. In addition, the prion domain of EWSR1 at the N-terminus provides the BAF complex with a gain of function that is recruited to GGAA repeats microsatellites, and when this occurs, the closed chromatin is remodeled for activation of enhancer and oncogenic transcription [33,34].

### 1.4. Intrinsically Disordered Proteins (IDPs)

Intrinsically disordered proteins (IDPs) and hybrid proteins containing ordered domains and intrinsically disordered protein regions (IDPRs) are functional proteins that play essential biological roles but do not have stable secondary or tertiary structure under physiological conditions [36]. IDPs and hybrid proteins that contain both intrinsically disordered and ordered parts have been found to be very common by computational analyses [37,38,39,40,41,42,43,44,45,46,47,48,49,50,51,52,53,54,55,56,57]. IDPs/IDPRs are characterized by exceptional structural heterogeneity. For example, extended IDPs/IDPRs possess a low content of hydrophobic amino acids, low sequence complexity (commonly contain repeats [58,59]), and have high levels of charged and polar residues [37,60,61,62,63,64,65,66]. Therefore, IDPs/IDPRs are characterized by the reduced informational content of their amino acid sequences, and their amino acid alphabet is decreased in comparison with the that utilized in the amino acid sequences of ordered proteins and domains [67]. Due to their unique amino acid biases, such proteins or regions are unable to fold into stable globular three-dimensional structures and exist as highly dynamic conformational ensembles consisting of multiple rapidly interchangeable structures [64,68,69,70,71,72,73,74]. The IDP structures can range from expanded, random coil-like conformations, to pre-molten globule-like ensembles, and to molten globular structures [70,72,75]. Some proteins are also characterized by the presence of semi-disorder; i.e., they have regions that possess ~50% predicted probability to be disordered or ordered [76]. Furthermore, different regions of a protein can be disordered to a different degree. As a result, IDPs/IDPRs are characterized by a very complex and heterogeneous spatiotemporal structural organization, possessing foldons (independent foldable units of a protein), inducible foldons (disordered regions that can fold at least in part due to the interaction with binding partners), non-foldons (non-foldable protein regions), semi-foldons (regions that are always in a semi-folded form), and unfoldons (ordered regions that have to undergo an order-to-disorder transition to become functional) [73,74,77,78].

Despite their lack of unique structures, IDPs/IDPRs play a number of crucial roles in regulation of signaling pathways and in the control and coordination of transcription, translation, and the cell cycle [68,75,79,80,81,82,83,84,85]. They also can be involved in cell protection, protein protection, cellular homeostasis, and controlled cell death [86,87,88,89]. Functional IDPRs are commonly found in enzymes [90]. Furthermore, IDPs play a central role in the assembly of large proteinaceous machines, such as ribosome, organization of chromatin, gathering of microfilaments and microtubules, transport through the nuclear pore, and in the binding and transport of small molecules [91,92,93,94,95,96]. It was also proven that IDPs/IDPRs are subjected to various post-translational modifications (PTMs) that serve as the most important means of disorder-centered regulation of protein functionality [97,98]. Furthermore, the mRNA regions affected by alternative splicing predominantly encodes IDPRs [99]. In addition, some proteins are capable of biological phase separation leading to the formation of various proteinaceous membrane-less organelles (PMLOs), which are commonly found in both prokaryotic and eukaryotic cells [100,101,102,103,104,105]. These proteins are characterized by structural/sequence modularity, invariably contain high levels of intrinsic disorder, and as a result of their highly disordered nature, are able to participate in weak multivalent interactions [84,106,107].

Based on their multifunctionality and highly polymorphic structural organization, one might expect that BAF complexes and their subunits should include high levels of functional intrinsic disorder. The goal of this study is to test this hypothesis via multifactorial bioinformatics analysis of the intrinsic disorder predisposition of various subunits of human BAF complex, combined with a multi-level computational research of the potential roles of intrinsic disorder in functionality of these subunits and their associations with the pathogenesis of various human diseases.

## 2. Results and Discussion

### 2.1. Global Disorder Analysis of the BAF Subunits

Since ATP-dependent chromatic remodeling complexes are characterized by highly polymorphic structures with variable subunit compositions and numerous accessory proteins, the sections below provide merely partial coverage of all the BAF subunits and accessory proteins. In fact, only the 30 major components of this complex are discussed here. The list of considered subunits include BRG1/SMARCA4, BRM/SMARCA2 BCL7A, BCL7B, BCL7C, BCL11A, BCL11B, BAF250A/ARID1A, BAF250B/ARID1B, BAF57/SMARCE1, BAF155/SMARCC1, BAF180/PBRM1, BAF200/ARID2, BRD7, BRD9, BAF60A/SMARCD1, BAF60B/SMARCD2, BAF60C/SMARCD3, BAF45A/PHF10, BAF45D/DPF2, BAF45B/DPF1, BAF45C/DPF3, BAF47/SMARCB1, SSXT/SS18, BAF170/SMARCC2, BAF53A/ACTL6A, BAF53B/ACTL6B, actin-β/ACTB, GLTSCR1/BICRA, and GLTSCR1L/BICRAL.

First, we analyzed the predicted percentage of intrinsic disorder (PPID) for each of the 30 subunits of human BAF complex. Figure 3A shows the 2D-disorder plot presenting the PPID_PONDR_^®^_VSL2_ vs. PPID_PONDR_^®^_VLXT_ plot to illustrate the peculiarities of disorder predisposition of these proteins. According to the overall levels of intrinsic disorder, proteins can be classified as highly ordered (PPID < 10%), moderately disordered (10% ≤ PPID < 30%), and highly disordered (PPID ≥ 30%) [108]. From this PPID-based classification and data shown in Figure 3A, it is clear that only three BAF subunits, BAF53A, BAF53B, and β-actin are moderately disordered proteins, with the remaining members of this family being highly disordered. Importantly, the vast majority of human BAF subunits are characterized by PPID exceeding 50%. These observations define a set of these proteins as one of the most disordered set of functionally related proteins.

To gain further insight into the nature of disorder in BAF subunits, we used charge-hydropathy cumulative distribution function (CH-CDF) analysis (see Figure 3B) [109,110,111], which is based on a combined utilization of two binary disorder classifiers (i.e., predictors that evaluate the predisposition of a given protein to be ordered or disordered as a whole): Charge-hydropathy (CH) plot [69,110] and cumulative distribution function (CDF) plot [110,112].

The primary difference between the binary predictors is that the CH-plot is a linear classifier that takes into account only two parameters of the particular sequence (charge and hydropathy), whereas CDF analysis is dependent on the output of the PONDR^®^ predictor, a nonlinear classifier, which was trained to distinguish order and disorder based on a significantly larger feature space. According to these methodological differences, the CH-plot analysis is predisposed to discriminate proteins with substantial amount of extended disorder (random coils and pre-molten globules) from proteins with compact conformations (molten globule-like and ordered globular proteins). On the other hand, CDF analysis discriminates all disordered conformations, including native coils, native pre-molten globules, and native molten globules, from ordered globular proteins. Therefore, this computational discrepancy provides a useful tool for discrimination of proteins with extended disorder from molten globules and hybrid proteins containing noticeable (comparable) levels of ordered and disordered regions. Here, positive and negative Y values in the CH-CDF plot correspond to proteins predicted within the CH-plot analysis to be extended or compact, respectively.

On the other hand, positive and negative X values are attributed to proteins predicted within the CDF analysis to be ordered or intrinsically disordered, respectively. Therefore, the CDF-CH phase space can be separated into four quadrants that correspond to the following expectations: Q1, ordered proteins; Q2, proteins predicted to be disordered by CDFs, but compact by CH-plots (i.e., putative native molten globules or hybrid proteins); Q3, proteins predicted to be disordered by both methods (i.e., proteins with extended disorder); and Q4, proteins predicted to be disordered by CH-plots, but ordered by CDF [109,110,111]. Figure 3B represents the results of this analysis and shows that the subunits of human BAF complex are located within the three quadrants within the CH-CDF phase space, Q1, Q2, and Q3. Based on this analysis, one can conclude that 27 proteins are expected to be mostly disordered, and only three human BAF subunits, actin-β/ACTB, BAF53A/ACTL6A, and BAF53B/ACTL6B, are mostly ordered.

Next, we analyzed the inter-complex interactability of BAF subunits using a publically available computational platform STRING, which integrates all the information on protein–protein interactions (PPIs), complements it with computational predictions, and returns the PPI network showing all possible PPIs of a query protein(s) [113]. Results of this analysis are shown in Figure 4A that represents BAF complex as a condensed and highly interconnected PPI network containing 29 nodes connected by 372 edges. In this network, the average node degree was 25.7, and the average local clustering coefficient (which defines how close its neighbors are to being a complete clique; the local clustering coefficient is equal to 1 if every neighbor connected to a given node *N_i_* is also connected to every other node within the neighborhood, and it is equal to 0 if no node that is connected to a given node *N_i_* connects to any other node that is connected to *N_i_*) was 0.981. Furthermore, since the expected number of interactions among proteins in a similar size set of proteins randomly selected from human proteome was equal to 43, the inter-BAF PPI network had significantly more interactions than expected, being characterized by a PPI enrichment *p*-value of <10^−16^. We also used STRING to study the engagement of the subunits of the BAF complex in interactions with 500 proteins forming the first shell of the resulting interactome (note that the number of interactors in STRING was limited to 500). In this analysis, the high confidence level of 0.7 was used. Figure 4B represents the resulting interactome that included 529 nodes connected by 11,679 edges. Therefore, this interactome was characterized by an average node degree of 44.2 and shows an average local clustering coefficient of 0.691. The expected number of interaction for the set of proteins of its size was 3606, indicating this BAF-centered PPI network had significantly more interactions than expected (PPI enrichment *p*-value was < 10^−16^).

### 2.2. Transcriptional Activator BRG1 or ATP-Dependent Helicase SMARCA4 (PPID = 61.3%)

Human Brahma-related gene-1 (BRG1, also known as BAF190A, or SWI/SNF-related matrix-associated actin-dependent regulator of chromatin subfamily A member 4 (ATP-dependent helicase SMARCA4), UniProt ID: P51532; 1,647 residues) encoded by the *SMARCA4* gene is the central catalytic subunit of numerous chromatin-modifying enzymatic complexes, including BAF [114]. BRG1 alone is able to promote some nucleosome remodeling in vitro, while almost optimal chromatin remodeling can be achieved by adding BAF170, BAF155, and BAF47 [115]. In addition to the BAF complex, BRG1 is found in many other complexes, including: The WINAC complex (WSTF including nucleosome assembly complex) sharing subunits with both SWI/SNF- and ISWI-based chromatin remodeling complexes [116]; the NUMAC complex (nucleosomal methylation activation complex) containing histone-modifying enzyme co-activator-associated arginine methyltransferase-1 (CARM1) in addition to BRG1 and several other SWI/SNF-associated proteins [117]; the NCoR-1 (nuclear receptor corepressors-1) complex including four core SWI/SNF proteins, BRG1, BAF170, BAF155, and BAF47, histone deacetylases-3 (HDAC3), and the transcriptional co-repressors KAP-1 that interacts with proteins possessing KRAB domain, which is a common motif found in DNA-binding transcriptional repressors [118]; and the mSin3A/HDAC complex, known to repress transcription and induce gene silencing [114].

BRG1 plays a role in cellular proliferation and differentiation by interacting with specific promoters [119], and serves as a tumor suppressor [120] or transcription activator or a transcription repressor via interaction with specific protein partners [114]. Among the most noticeable BRG1 partners related to the transcription activation are several nuclear receptors (NRs) such as AR, ERα, GR, PPARγ, PR, and VDR, as well as β-catenin, CARM1, EVI-1, Mef2D, p130(RN2), Samd3, STAT1, and STAT2 [114]. BRG1-controlled transcriptional repression depends on the BRG1 interaction with GR, HDAC1/2, HP1, Mbd3, Mi-2β, mSin3A, pRb, PRMT5, REST, SMRT, and SYT-SSX, whereas tumor suppression activity of BRG1 is achieved via its interaction with BRCA1, p53, and FANCA (which also act as transcription activators) [114].

The sections below briefly describe mutations in, and deregulations of, *SMARCA4*/BRG1 associated with the development of human diseases. Since this protein acts as a tumor suppressor interacting with key regulatory proteins such as BRCA1 and p53 [121,122,123], loss of its expression leads to abnormal proliferation, anomalous cell cycle control, and altered cell growth through a variety of ways [121,124]. Furthermore, mutations in BRG1 were found in 5–10% of cases of lung cancer [125,126]. In the case of non-small cell lung cancers, it was also established that patients with BRG1-negative lung cancer had worse prognoses than patients with BRG1-positive non-small cell lung cancer [17,121].

Small cell carcinoma of the ovary of hypercalcemic type (SCCOHT) is a very rare tumor usually found in young females [17]. Hypercalcemia is found in most cases, and is associated with a poor prognosis [127,128]. In almost all SCCOHT tumors, inactivating mutations leading to loss of BRG1/*SMARCA4* expression have been exclusively found [129], and mutations of this protein were observed in 91% of cases [125,126]. Blood neoplasm (Burkitt lymphoma) is linked to mutations in the BRG1*/SMARCA4* and BAF250A*/ARID1A* tumor suppressor genes [17,130], with the nonsense and frame-shift mutations leading to loss of function constituting almost 30% of the genetic events occurring with BAF250A*/ARID1A* [130]. Group 3 and 4 pediatric medulloblastomas also linked to mutation in the BRG1*/SMARCA4* gene represent a significant clinical challenge due to poor prognosis and lack of targeted therapy [17,131].

Pancreatic ductal cells undergo a stage of dedifferentiation before progressing into intraductal mucinous papillary neoplasia, after which they progress to pancreatic ductal adenocarcinoma, and during this step of progression BRG1 was found to play a dual role according to the stage of the tumor, acting either as a tumor suppressor or a promoter of the pancreatic tumorigenesis [17,132]. BRG1 acts as a tumor suppressor during the early stages of dedifferentiation of pancreatic ductal cells, and also promotes late stage progression of pancreatic ductal adenocarcinoma [132].

Rhabdoid tumors are rare and aggressive embryonic tumors that affect the central nervous system, and which are typically diagnosed in early childhood, usually under three years old [133]. A rhabdoid tumor can involve the kidney and other sites, particularly soft tissues [134]. These cancers have a poor prognosis, with a 5-year survival of 33%, as they are usually at stage 3 or even higher at the time of diagnosis [135]. It was shown that the bi-allelic BRG1 gene mutation is present in 100% of the tumors (total *n* = 4) [125,126].

The role of BRG1 in breast cancer pathogenesis is determined mostly by its interaction with a tumor suppressor BRCA1. Although mutated BRCA1 exposes individuals to the risk of breast cancer, it was also shown that BRCA1 could interact with the BRG1 subunit of a SWI/SNF complex. Furthermore, a mutant form of BRG1 interfered with the transcriptional control mediated by BRCA1 [123]. Other tumors associated with *BRG1* mutations are melanomas, colorectal cancer, oral cancer, and hepatocellular carcinoma, with BRG1 mutations being present in 5–10% of cases in melanoma and colorectal cancer, and increased expression of BRG1 mRNA identified in both oral cancer and hepatocellular carcinoma [125,126].

Another malady linked to the mutated BRG1 is Coffin–Siris syndrome (CSS), a disease characterized by growth and developmental abnormalities as well as craniofacial features, such as sparse scalp hair, bushy eye brows, wide and prominent mouth, body hirsutism, absent or hypoplastic nails of fifth fingers, or toes with absent hypoplastic phalanges. The disease is sometimes associated with cardiac anomalies like atrial septal defect and patent ductus arteriosus, and also gastro intestinal anomalies like pyloric stenosis and intestinal malrotation [136]. Chromatin remodeling subunit SMARCA4/BRG1 has been found to be mutated in 11% of CSS patients, containing missense mutations with gain-of-function or dominant-negative effects [129]. Besides CSS, other neurodevelopmental disorders, such as Hirschsprung disease, autism spectrum disorder, and schizophrenia, have been associated with mutations in BRG1/BRM subunit of the BAF chromatin remodeling complex like [137]. Furthermore, SMARCA4/BRG1 mutations have been also linked to microphthalmia, a developmental disorder in which one or both eyes are abnormally small and have anatomic malformations [129]. This linkage is defined by the fact that BRG1 is involved in lens development, and controls both terminal differentiation of lens fiber cells and organized degradation of their nuclei (denucleation) in embryos [129,138].

Multifunctionality of BRG1 and link of this protein to a broad spectrum of human diseases are determined by the complexity of its structural organization (see Figure 5). First, this protein has advanced domain organization, containing a series of conserved functional domains and motifs. Among these functional domains and motifs are QLQ domain (residues 171–206) involved in protein–protein interactions, DNA-binding HSA domain (residues 460–532), BRK domain (residues 611–656), helicase (residues 766–1246) containing DEXHc ATP binding domain (residues 766–931) and helicase C-terminal domain HELICc (residues 1081–1246), and bromodomain (residues 1455–1566) interacting with acetylated lysines (Figure 5A). BRG1 also contains the Snf2 ATP coupling domain (residues 1321–1389), and several established functional regions, such as residues 1–282, necessary for interaction with SS18L1/CREST, RNA binding region interacting with lncRNA Evf2 (residues 462–748), and two regions sufficient for interaction with DLX1 (residues 837–916, and 1247–1446). There are also several regions with compositional biases, such as the poly-Lys region (residues 578–588) and three poly-Glu regions (residues 663–672, 1360–1364, and 1571–1584). Structural information is currently available for bromodomain by itself (see Figure 5B) and for a fragment of C-tail (residues 1591–1602) complexed with the Brd3 ET domain (see Figure 5C). BRG1 is characterized by high intrinsic disorder content, with >60% residues of this protein being predicted to be disordered (Figure 5D). Furthermore, D^2^P^2^-generated disorder profile of this protein show BRG1 contains 27 molecular recognition features (MoRFs); i.e., IDPRs capable of undergoing induced folding at interaction with binding partners, as well numerous posttranslational modifications (PTMs).

Importantly, BRX1 C-terminal region complexed with the Brd3 ET domain (see Figure 5C) coincides with one of the C-terminally located MoRFs (residues 1592–1600). Similarly, the protein interacting QLQ domain (residues 171–206) is a part of a long N-terminal MoRF (residues 21–231). Similarly, the 34-residue-long MoRF (residues 475–508) represents a large portion of the HSA domain spanning residues 460–532. Overall, 638 residues of human BRG1 (38.7%) or ~60% of its predicted disordered residues are expected to be involved in disorder-based protein–protein or protein–nucleic acid interactions. Finally, BRG1 might exist in five experimentally validated isoforms generated by alternative splicing (AS), and there are 29 computationally mapped potential isoforms. Canonical BRG1 isoform is a 1647-residue long protein with highly disordered N- and C-terminal regions. Residues 1259–1291 are missing in four AS-generated isoforms. In addition, isoforms 3 and 4 have residue W_1,388_ changed to a tetrapeptide WLKT, and isoforms 3 and 5 are missing residue S_1,475_. Note that all AS-induced sequence changes are concentrated within the intrinsically disordered C-terminal region of BRG1. These AS events may also affect disorder-based functionality of this protein. In fact, the missing 1259–1291 region includes one of the MoRFs (residues 1263–1282), changes at W_1,388_ affect another MoRF (residues 1381–1398), and elimination of S_1,475_ might affect yet another MoRF (residues 1458–1474). STRING analysis of the BRG1 interactome (using the highest confidence of 0.9) produced a PPI network containing 77 nodes with an average node degree of 24.5 and average local clustering coefficient of 0.836. Instead of the expected 130 edges, this network contained 943 interactions (PPI enrichment *p*-value < 10^−16^). The corresponding BRG1-centerd PPI network is shown in Appendix A.

### 2.3. Protein Brahma Homolog BRM or Probable Global Transcription Activator SNF2L2 (PPID = 60.6%)

Human protein brahma homolog (BRM, also known as BAF190B, SWI/SNF-related matrix-associated actin-dependent regulator of chromatin subfamily A member 2 (ATP-dependent helicase SMARCA2), or probable global transcription activator SNF2L2, UniProt ID: P51531) is a 1590 residue-long catalytic subunit of the multiprotein chromatin-remodeling SWI/SNF complexes. BRM is encoded by the *SMARCA2* gene, and has domain organization similar to that of BRG1, containing a QLQ domain (residues 173–208), an HSA domain (residues 436–508), a helicase ATP-binding domain (residues 736–901), a helicase C-terminal domain (residues 1054–1216), and a bromodomain (residues 1419–1489). There are also several regions with the compositional bias in the amino acid sequence of human BAF190B. These include two poly-Gln regions (residues 216–238 and 245–253), a poly-Arg region (residues 559–562), and three poly-Glu regions (residues 643–650, 1291–1301, and 1518–1529).

Since BRM serves as an alternative to BRG1 in some chromatin-remodeling SWI/SNF complexes, and since these two proteins have overlapping functions, only some of the pathological consequences of BRM misbehavior as well as the structural and intrinsic disorder-related features of this protein was considered here. Multiple mutations in the *SMARCA2* gene were identified in Nicolaides–Baraitser syndrome (NCBRS), which is a rare disease characterized by severe mental retardation with absent or limited speech, seizures, short stature, sparse hair, typical facial characteristics, brachydactyly, prominent finger joints, and broad distal phalanges [139]. These *SMARCA2* gene mutations cause BRM substitutions at Ala_752_, Arg_755_, Ile_756_, His_851_, Asp_852_, Lys_852_, Arg_854_, Asn_854_, Gly_855_, Arg_881_, Val881, Leu_883_, Tyr_939_, Ser_946_, Phe_946_, Cys_1105_, Pro_1105_, Pro_1135_, Arg_1146_, Val_1158_, Gly_1159_, Leu_1159_, Gln_1159_, His_1162_, Pro_1188_, Val_1201_, Cys_1202_, Gly_1205_, and Trp_1213_. Single nucleotide polymorphisms (SNPs) in this gene are also associated with schizophrenia [140].

According to the BioMuta database of single-nucleotide variations (SNVs) in cancer (https://hive.biochemistry.gwu.edu/biomuta/), mutations in the *SMARCA2* gene are linked to melanoma, malignant glioma, thyroid carcinoma, and blastoma, as well as to uterine, urinary bladder, stomach, colorectal, kidney, ovarian, breast, liver, cervical, and lung cancer.

X-ray crystal structure was solved for a construct containing helicase ATP-binding domain (residues 705–955) with N-terminally attached maltose binding protein (MBP) tag (PDB ID: 6EG3) [141]. Figure 6A shows the structure of this ATP-binding domain with the computationally removed MBP tag. The NMR solution structure is also known for a bromodomain (residues 1377–1504; PDB ID: 2DAT; see Figure 6B). Figure 6C shows that similar to BRG1, N- and C-termini of BRM were predicted to be highly disordered and contain numerous PTMs sites and MoRFs. In fact, there were 23 MoRFs in this protein that span over 524 residues (32.9% of total residues or 54.4% of disordered residues). STRING-generated BRM-centered PPI network (using the highest confidence of 0.9) contains produced a PPI network containing 34 nodes with an average node degree of 19.5 and average local clustering coefficient of 0.842. Instead of the expected 39 interactions, this network contained 332 edges. Therefore, this PPI network was significantly more connected than expected (PPI enrichment *p*-value < 10^−16^). The corresponding BRM interactome is shown in Appendix A.

### 2.4. BAF155 (PPID = 59.2%)

BAF155 is a 1105 residue-long protein (UniProt ID: Q92922) encoded by the *SMARCC1* gene. BAF155 plays a critical role within SWI/SNF complex, being involved in keeping the whole BAF-complex stoichiometry [142,143]. This protein is a part of npBAF and nBAF complexes. BAF155 stimulates the ATPase activity of the catalytic subunit BRG1 of the BAF complex [18,115]. Together with the BRG1/BRM and BAF47, BAF155/BAF170 forms an evolutionarily conserved stable complex able to carry out essential chromatin remodeling activity. Therefore, these subunits constitute the core components of the BAF complex. There is also evidence that supports the notion that BAF155 controls the steady-state level of BAF57 [143], and an interplay between those subunit and proteasome–protein mediated degradation is involved in the regulatory process [143]. Furthermore, BAF155 has a tumor suppressor activity in some tumors, such as colorectal cancer and ovarian carcinoma, as mutations of *SMARCC1*/BAF155 were also linked to the development of these tumors [144,145]. Prostate cancer is one of the most common cancers world-wide and carries the sixth place as the leading cause of cancer related mortality among men [144].

A diagnosis of prostate cancer is usually based on digital rectal exam (DRE), level of PSA, and biopsy [146]. It was proven that BAF155 expression is lost in prostate cancer, and that this protein plays a role in prevention of proliferation of prostate cancer cells (PC3), prevents the cancerous cells migration, and initiates the cancerous cells apoptosis process [125,144].

Human BAF155 (UniProt ID: Q92922) contains two functional domains, SWIRM (residues 449–546) that mediates specific PPIs required for the assembly of chromatin–protein complexes, and SANT (residues 618–669), which is another PPI module that can bind to tails of histone proteins. It also includes a coiled-coil domain (residues 914–946). Furthermore, there are several regions with amino acid composition biases in this protein, such as poly-Pro (residues 329–336), Glu-rich (residues 769–863), poly-Ala (residues 867–878), and Pro-rich regions (residues 977–1105).

Structure is currently known for the SWIRM domain complexed with the with the repeat 1 (RPT1) domain of BAF47 (residues 183–289; PDB ID: 5GJK) [20] and for the SANT domain (residues 610–675; PDB ID: 1RYU) [147] (see Figure 7A,B). Figure 7C,D illustrate that BAF155 contained high levels of intrinsic disorder and that these IDPRs are related to the regulation of its functionality. In fact, BAF155 was heavily decorated with multiple sites of various PTMs and had 19 MoRFs. Overall, 312 residues of this protein (28.2%) were expected to be engaged in disorder-based interactions. This feature is likely to define high interactability of human BAF155. In fact, STRING-based analysis revealed that this protein was forming a well-developed and highly connected PPI network that included 63 nodes linked by 850 edges, with an average node degree of 27 and an average local clustering coefficient of 0.831. Since the expected number of edges for the network of its size was 102, the BAF155-centered network had significantly more interactions than expected (PPI enrichment *p*-value was < 10^−16^).

### 2.5. BAF170 (PPID = 63.1%)

Human BAF170 (UniProt ID: Q8TAQ2) is a 1214-residue-long protein encoded by the *SMARCC2* gene. It has a domain organization similar to that of BAF155, containing SWIRM and SANT domains (residues 424–521 and 596–647, respectively), as well as a coiled-coil domain (residues 907–934). It also has regions with amino acid composition biases, such as poly-Glu (residues 186–189), Glu-rich (residues 747–855), poly-Ala (residues 861–870), poly-Gln (residues 956–960), and Pro-rich regions (residues 961–1213). *SMARCC2* is one of the genes mutations that are associated with a rare and clinically and genetically heterogeneous disorder, Coffin–Siris syndrome. Furthermore, according to the BioMuta database of single-nucleotide variations (SNVs) in cancer, mutations in this gene are associated with liver, cancer, colorectal, stomach, lung, breast, and uterine cancers, as well as with melanoma and multiple other tumors.

In addition to the canonical isoform, human BAF170 has two AS-generated isoforms, in which residue Q_550_ is changed to QGRQVDADTKAGRKGKELDDLVPETAKGKPEL, and which are missing residues 1075–1189 (isoform 2) or 1075–1167 (isoform 3). Although no structural information is currently available for human BAF170, its disorder profile and the peculiarities of disorder-based functionality are similar to those of BAF155. This is evident from the comparison of Figure 7 and Figure 8. BAF170 has multiple PTM sites and includes 25 MoRFs that cover 464 residues, indicating that 38.2% of residues of this protein can be engaged in disorder-based interactions. STRING-based analysis using the highest confidence of 0.9 revealed that the human BAF170 protein is forming a well-developed and highly connected PPI network that includes 64 nodes linked by 842 edges, with the average node degree of 26.3 and the average local clustering coefficient of 0.849. Since the expected number of edges for the network of its size was 100, BAF170-centered network was characterized by significantly more interactions than expected (PPI enrichment *p*-value was < 10^−16^).

### 2.6. BAF47 (PPID = 35.8%)

BAF47 is a 385-residue-long SWI/SNF-related matrix-associated actin-dependent regulator of chromatin subfamily B member 1 (SMARCB1, UniProt ID: Q12824) encoded by the *SMARCB1* gene. BAF47 serves as a core component of BAF, PBAF, and related multiprotein chromatin-remodeling complexes. This protein stimulates BRG1 activity in vitro and is related to the CSF1 promoter activation. BAF47 is able to interact with double-stranded DNA and also binds to several human proteins, such as CEBPB (when not methylated) [148], PIH1D1 [149], PPP1R15A [150,151], MYK, and MAEL [152]. Furthermore, BAF47 is engaged in interaction with several viral proteins, such as human immunodeficiency virus-type 1 (HIV-1) integrase, human papillomavirus 18 E1 protein [153], and Epstein-Barr virus protein EBNA-2 [154]. Mutations in the *SMARCB1* gene are associated with Coffin–Siris syndrome 3 (CSS3), Schwannomatosis 1 (SWNTS1), rhabdoid tumor predisposition syndrome 1 (RTPS1), as well as familial multiple meningioma, familial rhabdoid tumor, and neurofibromatosis type 3. Furthermore, according to BioMuta (https://hive.biochemistry.gwu.edu/biomuta/branchview/), SNV mutations in *SMARCB1* are found in various cancers, including uterine, lung, kidney, and stomach cancers.

Human BAF47 can be divided into several functional regions with specific activities, such as DNA-binding region (residues 1–113), HIV-1 integrase binding region (residues 183–243) that overlaps with MYC-binding region (residues 186–245), and PPP1R15A-interacting region (residues 304–318). There are also two repeat regions (residues 186–245 and 259–319). NMR solution structures were solved for the DNA-binding domain (residues 2–113, PDB ID: 5AJ1) and for the MYC/HIV-1 integrase binding region (residues 183–265; PDB ID: 6AX5; see Figure 9A,B, respectively).

Although human BAF47 contains less disorder than other core subunits of the BAF complex, Figure 9C,D show that disorder is mostly concentrated within the N-terminal half of this protein, where one can find several PTM sites and three MoRFs. Interestingly, BAF47 is one of a few proteins in the set we analyzed that does not contain regions with amino acid sequence biases.

According to STRING analysis, where the highest confidence level of 0.9 was used, the PPI network formed by human BAF47 includes 58 nodes connected by 765 edges, with an average node degree of 26.4 and an average local clustering coefficient of 0.866. Since the expected number of edges for the network of its size was 89, BAF47-centered network possessed more interactions than expected (PPI enrichment *p*-value was < 10^−16^).

### 2.7. BAF250A (PPID = 75.8%)

The distinguishing subunit of human SWI/SNF is BAF250/ARID1, which exists as two forms, BAF250A/ARID1A and BAF250B/ARID1B [155] that enroll the SWI/SNF complex to chromatin, allowing transcriptional activation of several genes [156]. As an example, the BAF250A excites GR-dependent transcriptional activation, and the stimulation is sharply reduced when the C-terminal region of BAF250A is deleted. It is suggested that BAF250A may promote engagement of the human BAF-complex to its targets through either protein-DNA or protein–protein interactions [157]. Furthermore, BAF250A, which is considered to be a regulatory subunit of the SWI/SNF, plays a key role in cardiac progenitor cell differentiation [158]. BAF250A serves as one of the core subunits of the npBAF and nBAF complexes, which are neural progenitors-specific chromatin remodeling complex and the neuron-specific chromatin remodeling complex, respectively. Although the npBAF complex is essential for the self-renewal/proliferative capacity of the multipotent neural stem cells, nBAF regulates the activity of genes essential for dendrite growth.

Similar to BRG1, mutations and deregulation of BAF250A is associated with cancer pathogenesis. For example, Burkitt lymphoma has been linked to mutations in the genes coding for both BRG1 and BAF250A [17,130], with nonsense and frame-shift mutations in BAF250A that lead to loss of function constituting almost 30% of genetic events associated with this blood neoplasm [130]. Mutations in *ARID1A* encoding BAF250A associated with the loss of BAF250 function have been linked to ovarian clear cell carcinoma (OCCC) [17,125,127], which is a rare form of cancer, accounting for approximately 5–10% of all ovarian carcinomas in North America, with a higher percentage in East Asia. OCCC is usually associated with endometriosis and can by diagnosed at an earlier stage, resulting in an overall positive prognosis [159]. Mutations in *ARID1A* with loss of BAF250 function have been also linked to the development of the subtypes of endometrioid carcinoma [160,161]. In OCCC and endometrioid ovarian cancer, *ARID1A* mutations may be heterozygous or biallelic, with biallelic mutations being associated with BAF250A loss of function or loss of protein expression [125,126]. Mutations in *ARIDA1* were found in more than 10% of cases of OCCC [125,126]. *ARID1A* mutations are also present in more than 10% of liver cancer cases [125,126]. During initiation of the tumors, deletion of *ARID1A* was protective, causing a delay in the onset of hepatocellular carcinoma in mice, while it was also noticed that loss of *ARID1A* led to acceleration of colorectal cancer in mice. Based on these observations, it was concluded that the role of BAF250 in tumor progression is dependent on the nature of the disease [162,163]. In addition to the aforementioned pathologies, genome sequencing and comparative genomic hybridization (CGH) studies have detected *ARID1A* mutations or deletions in a share of additional cancer subtypes, such as gastric cancer (29%), breast cancer (4–13%), pancreatic cancer (33–45%), transitional cell carcinoma of the bladder (13%), Waldenström’s macroglobulinemia (17%) [164], and uterine cancer [125,126].

Human BAF250A is a 2285-residue-long protein (UniProt ID: O14497) containing AT-rich interaction domain (ARID, residues 1017–1108), with known NMR solution structure (see Figure 10A). ARID is a DNA-binding module present in numerous eukaryotic transcription factors involved in regulation of cell proliferation, differentiation and development. There are also four LXXL motifs (residues 295–299, 1709–1713, 1967–1971, and 2085–2089) and four regions of amino acid composition biases, including two poly-Glu regions (residues 479–482 and 561–567), a poly-Ser region (residues 998–1001), and a Glu-rich region (residues 1317–1404). In addition to the canonical form, BAF250 may exist in two AS-generated isoforms, with missing residues 1367–1583 (isoform-1) or 1–383 (isoform-2). Figure 10B shows that ~80% of BAF250A residues were predicted to be intrinsically disordered. Intrinsic disorder is preferentially concentrated within N-terminal 4/5 of the protein. According to the D^2^P^2^ profiling (Figure 10C), human BAF250A contained 38 MoRFs spanning in total 1416 residues (62%), with some MoRFs exceeding 100 residues in length. There were also numerous sites of different PTMs.

STRING analysis of the human BAF250A interactome using the highest confidence of 0.9 generated a PPI network containing 61 nodes with an average node degree of 27.1 and an average local clustering coefficient of 0.832. Since instead of the expected 102 edges, this network contained 828 interactions, BAF250A-centered interactome was found to be more connected than expected (PPI enrichment *p*-value < 10^−16^).

### 2.8. BAF250B (PPID = 76.9%)

Human BAF250B is a 2236-residue long protein (UniProt ID: Q8NFD5) encoded by the *ARID1B* gene. Similar to BAF250A, BAF250B serves as a core subunit of the npBAF and nBAF complexes, thereby playing a role in neural development, controlling the self-renewal/proliferative capacity of the multipotent NSCs and also regulating the activity of genes needed for dendrite growth.

It was also concluded that BAF250B may act as a tumor suppressor, thus controlling cellular growth and division, which, if lost, can lead to the development of a variety of tumors [165,166]. In line with this conclusion, *ARID1B* was found to be mutated in more than 10% of patients with OCCC [125,126]. *ARID1B* mutations were found in 5–10% of cases of colorectal cancer [125,126], which is the chief cause of morbidity and mortality all over the world, accounting for over 9% of all cancer incidence, and being the third most common cancer worldwide and the fourth most common cause of death [167]. Colorectal cancer patients usually present with rectal bleeding, anemia, abdominal pain, weight loss, anorexia, constipation, and positive fecal occult blood test [168]. *ARID1B* mutations are also found in 5–10% of cases of gastric cancer [125,126], which is one of the most common cancers and the second most common cause of cancer deaths worldwide. In most countries, the diagnosis of gastric cancers is made based on symptoms such as dysphagia, weight loss, palpable abdominal mass, vomiting, and gastro-intestinal bleeding [169].

Finally, *ARID1B* can be mutated in hepatocellular carcinoma (HCC) [125,126], which is a complex disease and a main cause of death in high endemic areas of hepatitis B virus (HBV) or hepatitis C virus (HCV) infection. Recently, the growing incidence of HCC in the west was observed due to an HCV epidemic and the increase in prevalence of chronic alcoholic liver disease. HCC is usually asymptomatic, but it should be suspected in patients with cirrhosis, when there is liver function worsening accompanied by the appearance of acute complications, such as ascites, encephalopathy, variceal bleed, jaundice, upper abdominal pain, and fever [170].

In addition to ARID domain (residues 1041–1159) human BAF250B contains two LXXL motifs (residues 419–423 and 2036–2040), a nuclear localization signal (residues 1358–1,77), and 14 regions with amino acid composition biases, such as two Ala-rich segments (residues 2–47 and 329–493), two Ser-rich fragments (residues 35-57 and 684-771), two Gln-rich regions (residues 107-131 and 574–633), two poly-Ser segments (residues 1034–1037 and 1441–1444), a His-rich (residues 81–114), a poly-Gln (residues 114–131), a poly-Ala (residues 932–935), and a poly-Pro region (residues 1833–1836), as well as 261- and 139-residue-long Gly-rich and Pro-rich regions (residues 141–401 and 1459–1597, respectively).

Figure 11 summarizes the structural and functional disorder-related information for human BAF250B. Similar to BAF250A, ARID domain (residues 1041–1159) of BAF250B was the only protein region with a known structure (see Figure 11A). BAF250B was predicted to contain more than 75% of disordered residues, mostly in the form of long IDPRs (Figure 11B). Figure 11C shows that there were 32 MoRFs in this protein that account for 1330 residues (59.5%), and IDPRs of BAF250B were heavily decorated with various PTMs. Several MoRFs coincided or overlapped with the regions characterized by amino acid composition biases, suggesting that such compositional biases might be related to MoRF-driven PPIs. There were four described AS-generated isoforms and 17 potential computationally mapped isoforms. In comparison with the canonical isoform, isoform-2 (2249 residues) included a change of Q_579_ to 14-mer peptide QDSGDATWKETFWL, isoform-3 (2289 residues) had a change of K_1,032_ to KDSYSSQGISQPPTPGNLPVPSPMSPSSASISSFHGDESDSISSPGWPKTPSSP, and isoform 4 (1486 residues) was missing residues 1–750. A comparison of the data summarized in Figure 11 with the description of alternative splicing-generated alterations in the BAF250B amino acid sequence indicates that AS might affect the potential disorder-based functionality of this protein. For example, isoform-4 was missing 11 MoRFs and multiple phosphorylation and ubiquitination sites, whereas sequence changes in isoform-2 might affect two MoRFs (residues 543–578 and 583–604), and isoform-3 might change the local phosphorylation predisposition of this protein and introduce a new MoRF. STRING analysis of the human BAF250B interactivity using the highest confidence of 0.9 resulted in a PPI network with 28 nodes, an average node degree of 19.8, and an average local clustering coefficient of 0.872. Since this network contained 277 interactions instead of anticipated 29, the resulting BAF250B interactome was found to be more connected than expected (PPI enrichment *p*-value < 10^−16^).

### 2.9. BAF57 (PPID = 77.6%)

Human BAF57 (encoded by the *SMARCE1* gene) is a 411-residue-long protein, which is present in all BAF (including the npBAF and nBAF complexes) and PBAF complexes of higher eukaryotes. This protein is able to interact with androgen and estrogen receptors, is able to regulate nuclear receptor function, and due to this ability, plays essential roles during steroid hormone responses as well as during the differentiation of skeletal muscle [171]. BAF57 is also involved in controlling the cell cycle through the regulation of the transcription of specific cell cycle-related genes [172]. A specific interaction of this protein with the CoREST co-repressor complex leads to the repression of neuronal specific gene promoters in non-neuronal cells [173]. *SMARCE1* mutations have been found in the germline of families with a history of meningiomas and sporadic tumors [125,174]. The spinal cord meningiomas grow from intradural attachments and then stretch the arachnoid covering them, causing pain and paraparesis [175]. Germline *SMARCE1* mutations were identified in 14% (9/63) of the solitary meningioma patients younger than 25 years, with all the affected individuals suffering from meningiomas of the clear cell type [174]. Heterozygous loss of function mutations was also found in 4/9 individuals and loss of protein was found in all tumors [125,126]. *SMARCE1* mutations are thought to be involved other types of cancer, including breast, ovarian, and prostate [174]. Furthermore, BAF57 plays a role as a prognosis marker in prostate cancer, where the overexpression of this protein is correlated with a higher tumor grade. This feature was identified as a poor prognostic factor in endometrial carcinoma as well [171,176,177].

Human BAF57 (UniProt ID: Q969G3) is a SWI/SNF-related matrix-associated actin-dependent regulator of chromatin subfamily E member 1, which is also known as BRG1-associated factor 57. No structural information is available for this protein, which contains a long coiled-coil region (residues 220–319), as well as Pro-rich and Glu-rich regions (residues 5–65 and 320–411, respectively). Figure 12A,B show that human BAF57 was a highly disordered protein with eight MoRF regions and multiple PTM sites. Alternative splicing generates multiple isoforms, which are different from the canonical isoform by alterations of highly disordered N- and C-terminal regions. This protein is a highly promiscuous binder known to interact with more than 120 partners. This observation is illustrated by Figure 12C showing the STRING-generated protein–protein interaction (PPI) network centered at human BAF57. This PPI network included 51 nodes connected by 667 edges. As a result, the average node degree was equal to 26.2, and the network was characterized by an average local clustering coefficient of 0.837.

### 2.10. BAF180 (PPID = 43.2%)

Protein polybromo-1 (hPB1, also known as BRG1-associated factor 180 or BAF180, UniProt ID: Q86U86) is a 1689-residue-long protein encoded by the *PBRM1* gene. BAF180 is required for stability of the PBAF protein and acts as a negative regulator of cell proliferation [178]. BAF180 acts as a tumor suppressor by functioning within the tumor suppressor pathways causing genetic inactivation in tumors [179], and also plays a role in tissue maintenance [180]. Mutations in the *PBRM1* gene are related to renal carcinoma [125]. Renal cell carcinoma accounts for 2–3% of all cancers, and are mostly prevalent in developed countries [181]. While many renal masses are asymptomatic, patients rarely present with palpable abdominal mass, flank pain, and gross hematuria [181]. A more common set of symptoms include hypertension, weight loss, cachexia, neuromyopathy, elevated ESR, hypercalcemia, polycythemia, and bone pain in case of metastasis [181]. Heterozygous or biallelic inactivating mutations were found in 41% of clear cell tumor cases [125,178] and mutations were found in more than 10% of renal cell carcinoma cases [125,126]. Epithelioid sarcoma (ES) is a high-grade rare malignancy that develops within soft tissue. It is known for its high rate of local recurrence, lymph node involvement, and distant metastasis [182].

The clinical picture of ES usually involves a slow growing, non-tender, and firm tumor that usually develop in hands, fingers, and forearms, and although the tumor usually starts as a single nodule, it is usually represented as multiple nodules at the time of diagnosis, appearing as tan white non-encapsulated nodules with infiltrating borders [182]. Loss of BAF180 protein expression was found in 83% of cases (*n* = 23) [125,183]. Intrahepatic cholangiocarcinoma is a subtype of a family of cholangiocarcinoma tumors that develop within the cholangiocytes of the biliary tree [184] and although they are considered rare, accounting for 20–25% of all cholangiocarcinomas, they are still the second most common primary liver cancer after hepatocellular carcinoma [184]. Patients usually present with nonspecific symptoms like abdominal pain, weight loss, and jaundice [184]. Mutations in *PBRM1* lead to loss of protein expression in 32% of cases of intrahepatic cholangiocarcinoma (*n* = 108) [125,185].

Human BAF180 includes eight functional domains, such as bromodomains 1 through to 6 (residues 64–134, 200–270, 400–470, 538–608, 676–746, and 792–862), as well as domains BAH1 and BAH2 (residues 956–1074 and 1156–1272). It also has a Pro-rich region (residues 1468–1599). Structure of bromodomains 1, 2, 3, 5, and 6 were solved (see Figure 13A–E). The overall disorder level of this protein was exceeding 43% (see Figure 13F,G), which defines BAF180 as a highly disordered protein. Figure 13F,G show that linkers between bromodomains were disordered and contained a large number of various PTMs. Fourteen of 21 MoRFs were also located within these linkers. The highest level of disorder was found in the C-terminal region (residues 1300–1689), which also included the remaining MoRFs and additional PTM sites. According to the STRING analysis conducted using the highest confidence level of 0.9, an interactome of human BAF180 included 28 nodes and 266 edges. Therefore, this PPI network had an average node degree of 19.0, and an average local clustering coefficient of 0.905. Since the expected number of edges for the network of its size was 30, the interactome of BAF180 was significantly more connected than expected (PPI enrichment *p*-value < 10^−16^).

### 2.11. BCL7A (PPID = 87.6%), BCL7B (PPID = 89.6%), and BCL7C (PPID = 90.0%)

B-cell CLL/lymphoma 7 protein family members A, B, and C (BCL7A, BCL7B, and BCL7C, UniProt IDs: Q4VC05, Q9BQE9, and Q8WUZ0, respectively) are relatively short proteins (of 210, 202, and 217 residues, respectively) encoded correspondingly by genes *BCL7A*, *BCL7B*, and *BCL7C*. BCL7A, BCL7B, and BCL7C were shown to serve as dedicated, non-exchangeable subunits of the SWI/SNF complex [126]. Human BCL7B can act as a positive regulator of apoptosis, playing a role in controlling the Wnt signaling pathway via negative regulation of the expression of Wnt signaling components CTNNB1 and HMGA1 [186]. It also can be related to the control of the cell cycle progression, maintenance of the nuclear structure, and stem cell differentiation [186], whereas human BCL7C may be involved in cell apoptosis and mitochondrial function [187].

Genetic aberrations of *BCL7A* (three-way translocation t(8;14;12)(q24.1;q32.3;q24.1) with MYC and immunoglobulin gene regions) are linked to B-cell non-Hodgkin lymphoma [188]. As per BioMuta annotations, among several tumors associated with non-synonymous SNVs in the *BCL7A* gene, the most abundant are uterine, hematologic, liver, and colorectal cancers. Mutations in the *BCL7B* gene can be related to lung tumor development or progression [186]. Aberrations of this gene are also linked to the Williams–Beuren syndrome (WBS) [189] and, according to BioMuta, to liver and uterine cancers. Furthermore, BCL7B is related to allergic reaction in human, e.g., serving as an IgE autoantigen in atopic dermatitis [190]. Alterations in the human *BCL7C* gene are associated with the Sézary syndrome, which is a leukemic form of cutaneous T-cell lymphoma [191]. *BCL7C* (together with several other genes encoding subunits of the SWI/SNF complex, such as *ACTB*, *ARID2*, *BCL11A*, *BCL11B*, *BCL7B*, *BRD7*, *DPF2*, *DPF3*, *SMARCB1*, *SMARCD1*, and *SS18L1*) was also linked to the development of alcoholism and alcohol-related problems [192].

Human BCL7C contains a Pro-rich region (residues 108–217), whereas there are no segments with the compositional biases in BCL7A and BCL7B proteins. All three BCL7 family members contain several isoforms generated by alternative splicing. In BCL7A, isoform 2 is extended relative to the canonical form containing Q → QVPRSRSQRGSQIGREPIGLSG substitution at position 187. Human BCL7B has four isoforms, where isoform 2 is missing residues 1–60 and has changed N-terminal region (residues 61–88), KSNSSAAREPNGFPSDASANSSLLLEFQ → MPGPWLCPEFLLRKMTTLSCCLCSVWFS, isoform 3 contains K → F substitution at position 163 and is missing residues 164–202, whereas residues 89–145 are missing in isoform 4. In BCL7C, isoform 2 is different from the canonical form by changed residues 177–217 (DSGVRMTRRALHEKGLKTEPLRRLLPRRGLRTNVRPSSMAVPDTRAPGGGS KAPRAPRTI PQGKGR). Furthermore, 104- and 293-residue-long potential isoforms are mapped to this protein.

Human BCL7A contains multiple phosphorylation and ubiquitination sites (see Figure 14A), as well as three MoRFs containing 159 residues. Since this corresponds to 75.7% of the entire sequence, the whole human BCL7A can be considered as a disorder-based interactor. Similarly, Figure 14B shows that highly disordered human BCL7B protein contains multiple PTM sites and five MoRFs (133 residues, 64.9%), whereas 154 residues of human BCL7C (71%) are involved in the formation of four MoRFs (see Figure 14C). The STRING-generated PPI network of human BCL7A using the highest confidence level of 0.9 contained 10 nodes and 39 edges, indicating that this PPI network had an average node degree of 7.6, and an average local clustering coefficient of 0.961. Since the expected number of edges for the network of its size was 9, the BCL7A interactome was significantly more connected than expected (PPI enrichment *p*-value < 10^−13^). The PPI network of human BCL7B generated by STRING using the highest confidence level of 0.9 contained eight nodes linked by 27 edges. The resulting interactome had an average node degree of 6.75 and an average local clustering coefficient of 0.964. This network had significantly more interactions than the expected seven edges (PPI enrichment *p*-value < 7.92 × 10^−9^). A comparable picture was observed for the PPI network centered at the human BCL7C, where STRING indicates the presence of nine nodes linked by 30 interactions, an average node degree of 6.67, and an average local clustering coefficient of 0.952. Since a random set of its size was expected to have eight edges, the BCL7C interactome was significantly more connected (PPI enrichment *p*-value < 2.44 × 10^−9^).

### 2.12. BCL11A (PPID = 69.0%) and BCL11B (PPID = 69.8%)

B-cell lymphoma/leukemia 11A and 11B (BCL11A and BCL11B; UniProt IDs: Q9H165 and Q9C0K0, respectively) are 835- and 894-residue long proteins encoded by BCL11A and BCL11B genes. Human BCL11A and BCL11B protein serve as transcription factors associated with the BAF complex, where they act as dedicated, non-exchangeable subunits [126]. BCL11A is involved in brain development [193] and serves as a repressor of fetal hemoglobin (HbF) [194]. Furthermore, it operates as B-cell and myeloid proto-oncogene, is required for B-cell formation in fetal liver, and plays an essential role in hematopoiesis, leukemogenesis, and lymphopoiesis. BCL11A may modulate the transcriptional repression activity of pituitary homeobox 2 protein (PITX2), also known as ALL1-responsive protein ARP1 [195]. The functional repertoire of BCL11B is different, since this protein is a transcription factor that acts as a tumor suppressor that represses transcription through direct, TFCOUP2-independent binding to a GC-rich response element [196]. BCL11B is involved in the processes of the development, differentiation, and survival of T-lymphocytes in the thymus, and serves as a regulator of IL2 promoter enhancing IL2 expression in activated CD4^+^ T-lymphocytes [197]. It participates in the differentiation of variable areas of the central nervous system, such as the hippocampal neurons and the medium spiny neurons of the striatum [198], acts as a regulator of the responsiveness of hematopoietic stem cells to chemotactic signals, and modulates expression of the CCR7 and CCR9 receptors directing the movement of progenitor cells from the bone marrow to the thymus [199]. BCL11B was also found to be highly expressed in the cells of the immune system and the skin epithelial cells, and is required for skin morphogenesis [198]. It participates in the regulation of terminal differentiation of keratinocytes and the development of the papillary structure of the lingual epithelium [198].

Due to the involvement of BCL11A in regulation of HbF, mutations in the human *BCL11A* gene can be associated with the pathogenesis of intellectual developmental disorder with persistence of fetal hemoglobin (IDPFH) [193]. Furthermore, according to the BioMuta database, mutations in this gene are related to a wide spectrum of tumors, such as uterine, lung, esophageal, stomach, hematologic, pancreatic, breast, colorectal, and liver cancers, as well as malignant glioma, blastoma, and melanoma. Mutations of *BCL11B* were found in 5.7–12.7% of cases of T-cell acute lymphoblastic leukemia (TALL) [125,126,196], which is responsible for almost 15% and 25% of acute lymphoblastic leukemia in pediatrics and adult cohorts, correspondingly [200]. TALL patients show high white blood cell counts and may present with organomegaly, especially mediastinal expansion and CNS involvement [200]. Patients may also present with weight loss, fever and bleeding (petechia or purpura), dyspnea, and fatigue, but the presentation within the pediatric group is usually limited to bone pain. A low blood cell count is usually evident on lab work [201]. Diagnosis is established by the presence of 20% or more lymphoblasts in the bone marrow or peripheral blood [202].

Human BCL11A protein contains a region required for nuclear body formation and recruitment of SUMO1 (residues 1–210), as well as Pro-rich and Glu-rich regions (residues 260–373 and 481–509, respectively). In human BCL11B, there are Glu- and Gly-rich regions (residues 529–553 and 569–661, respectively) and six zinc finger motifs of C_2_H_2_-type (residues 221–251, 427–454, 455–482, 796–823, 824–853, and 854–884).

Figure 15 shows that both proteins were predicted to be highly disordered. Structural information was currently available only for the short N-terminal peptide of human BCL11A bound to the histone-binding protein RBBP4 (see Figure 15A). This peptide is located within the disordered region in close proximity to MoRF. Both BCL11A and BCL11B had multiple sites of different PTMs and included multiple MoRFs.

High disorder levels and presence of multiple MoRFs contribute to the interactability of these two proteins. STRING-generated PPI network of human BCL11A using the high confidence level of 0.7 contained 14 nodes and 32 edges, indicating that this PPI network had an average node degree of 4.57, and an average local clustering coefficient of 0.85. Since the expected number of edges for the network of its size was 14, the BCL11A interactome was significantly more connected than expected (PPI enrichment *p*-value < 2.6 × 10^−5^). The PPI network of human BCL11B generated by STRING using the high confidence level of 0.7 contained 11 nodes linked by 18 edges. This network had an average node degree of 3.27 and an average local clustering coefficient of 0.865. This BCL11B interactome had significantly more interactions than expected 10 edges (PPI enrichment *p*-value was 0.0212).

### 2.13. SSXT (PPID = 84.1%)

Synovial sarcoma translocated to X chromosome protein (SSXT, also known as protein SYT or SS18; UniProt ID: Q15532) is a 418 residue-long protein encoded by the *SS18* gene. SS18 (SYT) is believed to act as a transcription co-activator through the interaction with unidentified sequence-specific DNA binding transcription factor [203]. Two isoforms generated by alternative splicing are engaged in nuclear receptor co-activation and general transcriptional co-activation. Isoform 2 is different from the canonical form of SSXT (isoform 1) since it is missing residues 295–325.

Aberrations in the *SS18* gene are linked to synovial sarcoma [125], which is an aggressive subtype of soft tissue sarcomas, and although it is named “synovial”, it arises from the mesenchyme. It usually affects young adults [204]. Tumors usually appear in extremities, with most of them being in the lower extremities, especially within 10 cm of the knee. Patients may present with a slow-growing, frequently painful tumor [204]. Synovial sarcoma is a gene fusion-driven cancer, in which the underlying event is the chromosomal translocation t(X,18; p11, q11) that creates fusion of SS18 to SSX1 and SSX2. SS18-SSX is present in almost 100% of synovial sarcomas [205]. In SS18-SSX, eight C-terminal amino acids of SS18 are replaced by 78 amino acids of the C-terminus of SSX1, SSX2, and, rarely, SSX4 [206]. Translocations of the *SS18* gene with *SSX1* or *SSX2* were found in 90% of the tumors (total *n* = 32) [125,207].

Human SSXT protein contain two imperfect tandem repeats of 13 amino acids (residues 344–369) in addition to a poly-Pro region (residues 95–99), and four SH2 binding motifs (residues 50–53, 374–377, 392–401, and 412–416). Region 2–186 is related to transcriptional activation, and the C-terminal region (residues 175–418) is Gln-rich. Figure 16 shows that the human SSXT protein was massively disordered, with 351 residues (84%) being involved in the formation of eight MoRFs. The PPI network of human SSXT generated using STRING with the highest confidence level of 0.9 included 25 nodes connected by 170 edges, indicating that this PPI network had an average node degree of 13.6, and an average local clustering coefficient of 0.861. Since the expected number of edges for the network of its size was 25, the SSXT interactome was significantly more connected than expected (PPI enrichment *p*-value < 10^−16^).

### 2.14. BAF200/ARID2 (PPID = 61.3%)

AT-rich interactive domain-containing protein 2 (ARID2 or BAF200; UniProt ID: Q68CP9) is a 1835 residue-long protein that is encoded by the *ARID2* gene and is involved in the stabilization of the chromatin remodeling complex SWI/SNF-B (PBAF). ARID2/BAF200 is thought to be a tumor suppressor gene. It also suggested that BAF200 is required for the proper migration and differentiation of subepicardial venous cells into arterial endothelial cells, supporting the notion that BAF200 has a role in embryogenesis and organ development as well as maintaining cellular identity [208].

Similar to *ARID1A* and *ARID1B*, mutations in the *ARID2* gene are linked to hepatocellular carcinoma [125]. Heterozygous inactivating mutations in the *ARID2* gene are found in 14% (total *n* = 43) of the HCV-associated tumors [125,209] and mutations were found in 5–10% of cases [125,126]. Aberrations in ARID were also observed in melanoma [125], which is a malignant tumor that affects the skin, particularly melanocytes. Melanomas can arise within the eyes and meninges and is responsible for 90% of deaths associated with cutaneous tumors worldwide [210].

There are four types of melanoma that are histologically distinct. The first melanoma type is the superficial spreading melanoma, which often presents with multiple colors and pale areas of regression with the possible development of secondary nodular areas. A characteristic histologic feature is the presence of an epidermal lateral component with pagetoid spread of clear malignant melanocytes throughout the epidermis [210]. The second type is nodular melanoma, which is primarily nodular, exophytic brown-black, often an eroded or bleeding tumor [210]. The third type is Lentigo maligna melanoma that arises often after many years from melanoma in situ, located mostly on the sun-damaged faces of older individuals. It is characterized histologically by a lentiginous proliferation of atypical melanocytes at the dermo-epidermal junction [210]. The final type is acral lentiginous melanoma, which is typically palmoplantar or subungual. It initially develops as an irregular, poorly circumscribed pigmentation, which is followed by a nodular region that mirrors the invasive growth pattern [210]. Heterozygous loss of function mutations in the *ARID2* gene is present in 7% of tumors (total *n* = 121) [125,211], and mutations were found in 5–10% of cases [125,126]. Another tumor commonly associated with the mutations in *ARID2* is lung cancer [125], which is the most diagnosed cancer and the chief cause of death in men from cancer, while for women it is the fourth most common cancer to be diagnosed and the second cause of death from cancer [212,213]. Lung cancer arises from the respiratory epithelial cells and is divided in to two main types, small cell lung cancer (SCLC), which is highly malignant and accounts for 15% of lung cancer cases, and non-small lung cancer, which accounts for the remaining 85% of cases [212,213]. Patients with lung cancer usually present with symptoms such as cough, dyspnea, pain, depressive symptoms, fatigue, and nausea/vomiting, with a variable intensity of the symptoms depending on the stage of the lung cancer [214]. In lung cancer, biallelic inactivating mutations in the *ARID2* gene were found in 5% in cases of non-small cell lung carcinoma (*n* = 82) [125,215]. Finally, *ARID2* mutations were found in 5–10% of the colorectal cancer cases [125,126].

Human BAF200 includes the ARID domain (residues 13–105), LXXLL motif (residues 313–317), two ARM repeats (residues 163–207 and 263–454), Gln-rich region (residues 793–1128), and zinc-finger motif of C2H2 type (residues 1632–1657). Figure 17 shows that the 500 N-terminal residues of this protein were mostly ordered, whereas the remaining sequence of human BAF200 was expected to contain high levels of disorder. This disordered part of BAF200 had multiple PTMs and several MoRFs. Note that region 1784–1835 that includes one MoRF (residues 1812–1818) was missing in isoform 2.

The PPI network of human BAF200 generated using STRING with the highest confidence level of 0.9 contained 25 nodes connected by 241 edges. This PPI network had an average node degree of 19.3, and an average local clustering coefficient of 0.908. The expected number of edges for the network of its size was 26, indicating that the BAF200-centerd PPI network was significantly more connected than expected (PPI enrichment *p*-value < 10^−16^).

### 2.15. BRD7 (PPID = 61.2%) and BRD9 (PPID = 60.1%)

Bromodomain-containing proteins 7 and 9 (BRD7, UniProt ID: Q9NPI1; and BRD9, UniProt ID: Q9H8M2) are 651- and 597-residue-long proteins encoded by *BRD7* and *BRD9* genes, respectively. Both BRD7 and BRD9 are components of the SWI/SNF chromatin remodeling subcomplex GBAF and play a role in chromatin remodeling and regulation of transcription [216,217]. Although they are characterized by 37.1% sequence identity (53.0% sequence similarity), and although they both contain bromodomains (residues 148–218 and 153–223 in BRD7 and BRD9, respectively) with very similar structural organizations, these two proteins have rather different functions. In fact, BRD7 acts as a coactivator for the p53-mediated transcriptional activation of a set of target genes, but also may act as transcriptional corepressor that down-regulates the expression of target genes. It can bind to the ESR1 promoter and recruit BRCA1 and POU2F1 to this promoter. It also can bind to acetylated histone H3 [218]. BRD7 also operates as an activator of the Wnt signaling pathway [219], is required for the p53-mediated cell-cycle arrest in response to oncogene activation [220,221], and can inhibit cell cycle progression from G1 to S phase [218]. It can negatively regulate the GSK3B phosphotransferase activity, induce GSK3B dephosphorylation [219], and promote p53 acetylation [220]. On the other hand, BRD9 can act as a chromatin reader interacting with acylated, acetylated, and butyrylated histones [216].

According to BioMuta, non-synonymous single-nucleotide variations (nsSNVs) in the *BRD7* gene are linked to bladder, breast, colorectal, liver, lung, urinary, and uterine cancers, as well as melanoma. No related information was available for BRD9, which is however known as rhabdomyosarcoma antigen MU-RMS-40, indicating a relation of this protein to rhabdomyosarcoma.

Structural information was available for the bromodomains of both proteins (~70 residues). These domains are structurally identical to the bromodomains of other proteins from the BAF complex (e.g., BRG1, BRM, and BAF180, see above).

In addition to bromodomain (residues 148–218) BRD7 contains a Lys-rich region (residues 3–91), a nuclear localization signal (residues 65–96), and a coiled coil domain (residues 536–567). BRD9 also possesses a Lys-rich region (residues 64–95) and bromodomain (residues 153–223). Although BRD9 does not have a nuclear localization signal or a coiled coil domain, it contains a recognition region for interaction with acylated and butyrylated residues of histone tails [216]. Figure 18 shows that both proteins were expected to contain high levels of disorder with the noticeable exception for their bromodomains. Furthermore, BRD7 and BRD9 were heavily decorated by various PTMs and had numerous MoRFs.

The STRING-generated PPI network of human BRD9 using the highest confidence level of 0.9 contained 34 nodes connected by 284 edges. This PPI network had an average node degree of 16.7, and an average local clustering coefficient of 0.953. The expected number of edges for the network of its size was 43, indicating that BCR7 interactome was significantly more connected than expected (PPI enrichment *p*-value < 10^−16^). The STRING-based PPI network of human BRD9 was less developed, containing just four nodes if the highest confidence level of 0.9 was used. However, using the less stringent confidence level of 0.7 generated a network with 11 nodes and 46 edges. This interactome had an average node degree of 8.36 and an average local clustering coefficient of 0.982. Since the expected number of edges for the network of its size was 10, the BRD9-centered PPI network had significantly more interactions than expected (PPI enrichment *p*-value was < 2.22 × 10^−16^).

### 2.16. BAF60A/SMARCD1 (PPID = 43.8%), BAF60B/SMARCD2 (PPID = 49.3%), and BAF60C/SMARCD3 (PPID = 50.5%)

SWI/SNF-related matrix-associated actin-dependent regulator of chromatin subfamily D members 1, 2, and 3 (SMARCD1 (UniProt ID: Q96GM5), SMARCD2 (UniProt ID: Q92925), and SMARCD3 (UniProt ID: Q6STE5), also known as BAF60A, BAF60B, and BAF60C, respectively) are proteins with length 515, 531, and 483 encoded by *SMARCD1*, *SMARCD2*, and *SMARCD3* genes. These three proteins serve as subunits of npBAF and nBAF complexes. BAF60A affects vitamin D-mediated transcriptional activity from an enhancer vitamin D receptor element (VDRE) and may serve as a link between mammalian BAF complexes and the vitamin D receptor (VDR) [222]. It also facilitates critical communication between nuclear receptors and the BRG1/SMARCA4 chromatin-remodeling complex for transactivation [223]. BAF60B controls granulocytopoiesis and the expression of genes involved in neutrophil granule formation, thereby serving as crucial regulator of myeloid differentiation [224]. Finally, BAF60C co-regulates gene expression with BRG1 in cardiac precursors, thereby participating in directing temporal gene expression programs in cardiogenesis [14].

As per BioMuta, non-synonymous single-nucleotide variations (nsSNVs) in the *SMARCD1* gene are linked to colorectal, liver, and uterine cancer. nsSNVs in *SMARCD2* can be associated with melanoma and liver, colorectal, stomach, and uterine cancers, whereas mutations in *SMARCD3* can cause liver, uterine and colorectal cancer, or melanoma.

The common structural feature of these three proteins is the presence of the SWIB domain (residues 291–390, 307–382, and 259–334 in BAF60A, BAF60B, and BAF60C, respectively). Furthermore, BAF60A contains a region for interaction with ESR1, NR1H4, NR3C1, PGR, and SMARCA4 (residues 43–167) [223], a region needed for interaction with SMARCC1 and SMARCC2 (residues 168–474) [223], and a region necessary for GR/NR3C1-mediated remodeling and transcription from chromatin and also required for GR/NR3C1 interaction with the BRG1/SMARCA4 complex in vivo. Although human BAF60A contains a poly-Lys region (residues 124–127), and although there is a long Pro-rich segment (residues 10–129) in BAF60B, human BAF60C includes no regions with amino acid composition bias. The AS-generated isoform 1 of BAF60A is missing residues 423–463. Residues 10–129 of human BAF60B represent a Pro-rich region. Figure 19 shows that all three proteins are expected to contain close to 50% disordered residues, which are preferentially concentrated within their N-terminal halves. Here, numerous sites of various PTMs are also found together with multiple MoRFs.

The STRING PPI network of human BAF60A generated using the highest confidence level of 0.9 contained 62 nodes linked by 832 edges. This interactome had an average node degree of 26.8 and an average local clustering coefficient of 0.852. Since the expected number of edges for the network of its size was 97, the BAF60A -centered PPI network had significantly more interactions than expected (PPI enrichment *p*-value was < 10^−16^). The STRING-generated PPI network of human BAF60B using the highest confidence level of 0.9 contained 29 nodes connected by 301 edges. This PPI network had an average node degree of 20.8, and an average local clustering coefficient of 0.689. The expected number of edges for the network of its size was 30, indicating that BAF60B interactome was significantly more connected than expected (PPI enrichment *p*-value < 10^−16^). The STRING PPI network of human BAF60C generated with the highest confidence level of 0.9 included 84 nodes and 1056 edges. This interactome had an average node degree of 25.2 and an average local clustering coefficient of 0.849. Since the expected number of edges for the network of its size was 115, the BAF60C-centered PPI network had significantly more interactions than expected (PPI enrichment *p*-value was < 10^−16^).

### 2.17. PHF10/BAF45A (PPID = 53.3%), DPF1/BAF45B (PPID = 35.8%), DPF2/BAF45D (PPID = 59.7%), and DPF3/BAF45C (PPID = 50.5%)

BAF45A, BAF45B, BAF45C, and BAF45D are important constituent BAF complexes harboring two plant homeodomain (PHD)-type zinc finger domains. PHD is a ~60-residues-long domain found in more than 400 eukaryotic proteins, mostly related to the chromatin structure maintenance and the chromatin-mediated transcriptional regulation [225].

PHD finger protein 10 (PHF10 or BAF45A (UniProt ID: Q8WUB8) is a 498 residue-long subunit of the npBAF complex encoded by the *PHF10* gene. This protein contains two zinc-finger motifs of the PHD type (residues 379–436 and 438–481), as well as a SAY domain crucial for transcription activation (residues 89–295), and two regions essential for the induction of neural progenitor proliferation (residues 89–185, and 292–334). One of the AS-generated isoforms of PHF10 is missing residues 1–47. Mutations in the *PHF10* gene are associated with uterine cancer and melanoma.

D4, zinc, and double PHD fingers family members 1, 2, and 3 (DPF1, DPF2, and DPF3) also known as BAF45B or zinc finger protein neuro-d4 (UniProt ID: Q92782), BAF45D or zinc finger protein ubi-d4 (UniProt ID: Q92785), and BAF45C or zinc finger protein DPF3 (UniProt ID: Q92784) are 380-, 391-, and 378-residue-long proteins encoded by *DPF1*, *DPF2*, and *DPF3* genes. The DPF1/BAF45B protein is important for developing neurons, acting as a subunit of the npBAF complex, participating in the regulation of cell survival, and also possibly functioning as a neurospecific transcription factor. In AS-produced isoforms 2 and 3, residues 321–330 are missing and residue K_226_ is changed to ICGKRYK NRPGLSYHYTHTHLAEEEGEENAERHALPFHRKNNHKQ, and in addition, isoform 3 is missing residues 1–82. Similar to PHF10/BAF45A, DPF1/BAF45B contains two zinc-finger motifs of the PHD type (residues 254–311 and 308–368). Mutations in the *DPF1* gene are linked to the pathogenesis of uterine, liver, and lung cancers.

DPF2/BAF45D binds modified histones H3 and H4, thereby playing an important role in transcriptional regulation [226,227]. In hematopoietic progenitor cells, it can also act as a negative regulator of myeloid differentiation [226]. DPF2/BAF45D can be related to the regulation of non-canonical NF-kappa-B pathway [228] and takes a part in the regulation of the development and maturation of lymphoid cells. nsSNVs in the *DPF2* gene are linked to Coffin–Siris syndrome 7 (CSS7) [229], melanoma, blastoma, and liver, lung, and uterine cancers. This protein has a zinc-finger domain of C2H2 type (residues 209–232) and two zinc-finger motifs of the PHD type (residues 270–330 and 327–377). It has an AS-generated isoform with missing residues 156–339. Structural information is available for both, C2H2-type zinc finger domain (Figure 20A) [230] and double PHD finger domain of human DPF2/BAF45D (Figure 20B) [227].

DPF3/BAF45C is a subunit of the nBAF complex and a muscle-specific component of the BAF complex. Being able to specifically bind to acetylated and methylated lysines on histone 3 and 4, this epigenetic key factor for heart and muscle development serves as histone acetylation and methylation reader [229]. Therefore, DPF3/BAF45C operates as a tissue-specific anchor between histone acetylations and methylations and chromatin remodeling [229]. nsSNVs in the *DPF3* gene are associated with blastoma, melanoma, stomach, liver, lung, and uterine cancer. Similar to DPF2/BAF45D, this protein contains a zinc-finger domain of C2H2 type (residues 198–221) and two zinc-finger motifs of the PHD type (residues 259–319 and 316–366).

AS-generated isoform-2 of DPF3/BAF45C lacks PHD-type zinc fingers and does not bind to acetylated histones H3 and H4, whereas in isoform-4, the 177–378 region is changed and shortened to RCPLPSLHCFLPSLCRDRC.

Several structures are available for the double PHD fingers of human transcriptional protein DPF3/BAF45C, including NMR solution structures of this 261–372 region bound to a histone H3 peptide acetylated at lysine 14, or to a histone H4 peptide containing acetylation at lysine 16, or to a histone H4 peptide containing N-terminal acetylation at serine 1 [231]. Figure 20C,D represent the intricately intertwined structures of these complexes, with a complex between DPF3/BAF45C and histone H4 peptide containing N-terminal acetylation at serine 1 serving as an illustration of “cloud binding”, where the histone H4 peptide remains mostly unstructured and highly dynamic in its DPF3/BAF45C-bound state. Results of the evaluation of functional intrinsic disorder in human BAF45A/PHF10, BAF45B/DPF1, BAF45D/DPF2, and BAF45C/DPF3 proteins are shown in Figure 21. These four proteins belong to the category of highly disordered proteins, possessing high overall levels of intrinsic disorder, containing multiple PTM sites and several MoRFs, the number of which ranges from 1 in BAF45B/DPF1 to 2 in BAF45C/DPF3, 5 in BAF45D/DPF2, and 6 in BAF45A/PHF10.

According to STRING analysis using the highest confidence level of 0.9, the PHF10/BAF45A protein was involved in a rather well-developed PPI network with 19 nodes, 159 edges, an average node degree of 16.7, and an average local clustering coefficient of 0.939. This interactome had significantly more interactions than the expected 19 edges for the network of its size (PPI enrichment *p*-value was < 10^−16^). In the STRING-generated PPI network of DPF1/BAF45B that was created using the high confidence level of 0.7, there were 18 nodes with 144 connections, which was significantly larger than the expected 18 edges for the network of this (PPI enrichment *p*-value was < 10^−16^). In this PPI network, an average local clustering coefficient was 0.959, and an average node degree was 16. The DPF2/BAF45D-centered interactome generated by the STRING analysis using the highest confidence level of 0.9 contained 21 nodes linked by 131 interactions. This PPI network was characterized by an average node degree of 12.5, and an average local clustering coefficient of 0.899, containing significantly more interactions than the expected 21 edges for the network of its size (PPI enrichment *p*-value was < 10^−16^). Finally, when interactability of human DPF3/BAF45C was analyzed by STRING using the highest confidence level of 0.9, the resulting PPI network contained 21 nodes connected by 169 edges, which indicates that this network had significantly more interactions than the expected 21 nodes (the corresponding PPI enrichment *p*-value was < 10^−16^). This DPF3/BAF45C-centered PPI network was characterized by an average node degree of 16.1 and an average local clustering coefficient of 0.909.

### 2.18. BAF53A/ACTL6A (PPID = 23.5%), BAF53B/ACTL6B (PPID = 25.4%), and β-Actin/ACTB (PPID = 22.1%)

This group includes three proteins with the least amount of intrinsic disorder. These proteins are actin-like protein 6A (UniProt ID: O96019; 429 residues), actin-like protein 6B (UniProt ID: O94805; 426 residues), and β-actin (UniProt ID: P60709; 375 residues). They are encoded by *ACTL6A*, *ACTL6B*, and *ACTB* genes, respectively. These proteins share a noticeable sequence similarity. In fact, 84.51% residues are identical in BAF53A/ACTL6A and BAF53B/ACTL6B, whereas the sequence identity between BAF53A/ACTL6A and β-Actin/ACTB and between BAF53A/ACTL6A and β-Actin/ACTB is equal to 37.33% and 38.34%, respectively. These proteins have no regions with compositional biases.

ACTL6A, also known as 53 kDa BRG1-associated factor A (BAF53A), is a component of various complexes with chromatin remodeling and histone acetyltransferase activity, such as the NuA4 histone acetyltransferase complex [232,233,234,235] that interacts with MYC and the adenovirus E1A protein [236], a NuA4-related complex [232,236], BAF [21,229], PBAF [237], and related multiprotein chromatin-remodeling SWI/SNF complexes (e.g., BAF53 [238] or neural progenitors-specific chromatin remodeling complex (npBAF complex), as well as INO80 complex INO80 [239,240,241,242]. Within the BAF complexes, BAF53A/ACTL6A interacts with SMARCA4/BRG1/BAF190A, and PHF10/BAF45A [243]. ACTL6A mutations have been found in patients with intellectual disability of variable severity, developmental delay, dysmorphic features, and digit abnormalities. Additional features may include genitourinary and cardiac defects. The disease phenotype resembles the Coffin–Siris syndrome and brachymorphism–onychodysplasia–dysphalangism syndrome [243]. Based on these observations it was concluded that the mutation-driven impairment of the human ACTL6A binding to β-actin and BRG1 affects the integrity of the BAF complex, and that the analysis for the presence of mutations in human *ACTL6A* gene should be considered in patients with intellectual disability, learning disabilities, or developmental language disorder [243]. In the AS-generated isoform-2 of BAF53A/ACTL6A the 1–42 region is missing.

ACTL6B, or 53 kDa BRG1-associated factor B (BAF53B), also serves as important component of BAF [237,244], PBAF [237,244], and related multiprotein chromatin-remodeling complexes (e.g., the polymorphic neuron-specific chromatin remodeling complex nBAF). According to the BioMuta database, non-synonymous single-nucleotide variations (nsSNVs) in the *ACTL6B* gene were found in liver, lung, and uterine cancers, and this gene was significantly down-regulated in esophageal cancer.

Actin, cytoplasmic 1, or β-actin, is a crucial and highly conserved constituent of cytoskeleton, where it polymerizes to produce filaments in the form of a two-stranded helixes further forming cross-linked networks in the cellular cytoplasm [245]. Therefore, this protein exists in monomeric (G-actin) or polymeric (F-actin) forms [245]. Human β-actin interacts with CPNE1 and CPNE4. This protein was identified in IGF2BP1-dependent mRNP granule containing untranslated mRNAs. In relation to the subject of this paper, human actin serves as an important component of the BAF complexes [12]. Actin is found in complexes with XPO6, Ran, ACTB, and PFN1, and can interact with DHX9, EMD, ERBB2, GCSAM, TBC1D21, and XPO6. Actin interaction with nuclear DNA helicase II (NDH II), also designated RNA helicase A, is direct and promotes attachment of nuclear ribonucleoprotein complexes to filamentous actin [246]. Actin can also bind to a tumor suppressor, down-regulated in renal cell carcinoma 1 (DRR1) [247,248]. Since actin is one of the most abundant, if not the most abundant eukaryotic protein, it is not surprising that mutations in *ACTB* genes are linked to numerous diseases, such as dystonia, juvenile-onset (DJO) [249], Baraitser–Winter syndrome 1 (BRWS1) [250], inherited myopathy [251], combined and complex dystonia [252], developmental malformations–deafness–dystonia syndrome, as well as multiple tumors, such as melanoma, blastoma, and hematologic, lung, urinary, uterine, bladder, kidney, and colorectal cancers. Furthermore, human *ACTB* genes are down-regulated in prostate and urinary bladder cancers, and up-regulated in breast and kidney cancers.

Aberrations in F-actin polymerization were also linked to intellectual and developmental disabilities (IDD) and autism spectrum disorders (ASD) [253]. Despite the exceptional importance of human β-actin, no high resolution structural information is available for this protein as of yet. In fact, although structure of F-actin complexed with several partners was resolved by cryo-electron microscopy, only a short peptide of actin (residues 66–88) was co-crystallized with the binding partner, human histone-lysine N-methyltransferase SETD3 (see Figure 22A).

This is not very surprising, considering that actin is characterized by a thermodynamically unstable quasi-stationary structure with elements of intrinsic disorder [254,255]. Since structural and functional peculiarities of this most abundant protein of the eukaryotic cell were systematically analyzed in several recent reviews, we will not provide further details here, and direct the interested reader to those articles [254,255]. Figure 22B–D provide a glance on the functional intrinsic disorder of human BAF53A/ACTL6A, BAF53B/ACTL6B, and β-actin/ACTB and show the presence of relatively short IDPRs. BAF53A/ACTL6A and BAF53B/ACTL6B have short MoRFs, but there are no sites of intrinsic disorder-based interactions in β-actin/ACTB. On the other hand, human β-actin/ACTB contains a very large number of PTMs. It seems that this protein is the most modified component of the BAX complex.

In the STRING-generated PPI network of human BAF53B/ACTL6B that was created using the highest confidence level of 0.9, there were 98 nodes with 1,422 connections, which was significantly larger than the expected 184 edges for the network of its size (PPI enrichment *p*-value was < 10^−16^). In this PPI network, an average local clustering coefficient was 0.804, and an average node degree was 29. In BAF53B/ACTL6B-centered PPI interaction network generated by STRING with the highest confidence level of 0.9 there were 28 nodes linked by 294 interactions, which indicates that this network had significantly more interactions than the expected 29 nodes (the corresponding PPI enrichment *p*-value was < 10^−16^). This interactome was characterized by an average node degree of 21 and an average local clustering coefficient of 0.877. According to the STRING analysis using the highest confidence level of 0.9, human β-actin/ACTB was involved in an extremely well-developed and dense PPI network containing 204 nodes connected by 3086 edges. This network was characterized by an average node degree of 30.3 and an average local clustering coefficient of 0.896. This interactome had significantly more interactions than the expected 717 edges for the network of its size (PPI enrichment *p*-value was < 10^−16^).

### 2.19. GLTSCR1/BICRA (PPID = 88.2%) and GLTSCR1L/BICRAL (PPID = 74.8%)

BRD4-interacting chromatin-remodeling complex-associated protein (BICRA), also known as glioma tumor suppressor candidate region gene 1 protein GLTSCR1 (UniProt ID: Q9NZM4), is a 1560-residue-long component of the SWI/SNF chromatin remodeling subcomplex GBAF and an ncBAF complex subunit encoded by the *BICRA* gene.

GLTSCR1/BICRA is shown to play a role in BRD4-mediated gene transcription via interaction with the extraterminal (ET) domain of bromodomain protein 4 (BRD4) [256] that is important for the organism development, cancer progression, and virus-host pathogenesis [257]. Furthermore, GLTSCR1/BICRA also binds to NSD3, along with several components of the BRG1-CREST complex, thereby linking BRD4 to a chromatin remodeling complex [256]. Alterations (nsSNVs) in the *BICRA* gene are associated with various tumors, such as melanoma, blastoma, and thyroid carcinoma, as well as stomach, breast, liver, lung, colorectal, and uterine cancers. Furthermore, the *BICRA* gene is significantly overexpressed in urinary bladder cancer, lung squamous cell carcinoma, breast cancer, and kidney cancer. GLTSCR1/BICRA is an important subunit of the ncBAF complex critical for rhabdoid tumor cell line survival suggesting that GLTSCR1 inhibition can serve as a promising target for rhabdoid tumor treatment [258].

Figure 23 shows that human GLTSCR1/BICRA is one of the most disordered subunits of BAF complexes, possessing ~90% disordered residues. This protein contains several PTM sites and 32 MoRFs covering 944 residues.

This indicates that 60.5% of the total residues or 68.6% of the disordered residues in this protein can be engaged in disorder-based interactions. There are five regions with compositional bias in human GLTSCR1/BICRA, which are mostly concentrated in the C-terminal half of the protein. These include three poly-Pro segments (residues 935–940, 1333–1337, and 1345–1355), a poly-Gly region (residues 88–96), and a poly-Ser region (residues 1265–1276).

In a PPI network of human GLTSCR1/BICRA generated using STRING with high confidence level of 0.7, there were eight nodes and 20 edges (instead of seven expected edges). This network was characterized by an average node degree of 5, and an average local clustering coefficient of 0.869, containing significantly more interactions than expected for the network of its size (PPI enrichment *p*-value was < 4.94 × 10^−5^).

BRD4-interacting chromatin-remodeling complex-associated protein-like (BICRAL) also known as glioma tumor suppressor candidate region gene 1 protein-like (GLTSCR1; UniProt ID: Q6AI39), is a 1079-residue-long component of the SWI/SNF chromatin remodeling subcomplex GBAF encoded by the *BICRAL* gene. The currently available information about this protein is very limited. It is known that GLTSCR and GLTSCR1L are mutually exclusive paralogs incorporated in chromatin remodeling subcomplex GBAF, where they are used instead of an ARID protein [18]. It was also shown that *GLTSCR1* or *GLTSCR1L* knockouts in the metastatic prostate cancer cell line PC3 led to the loss of proliferation and colony-forming abilities [18].

Figure 24 represents the D^2^P^2^-generated profile of functional disorder in human GLTSCR1L/BICRAL and shows that similar to its paralog, GLTSCR1/BICRA, this protein was predicted to have high levels of intrinsic disorder, contained multiple PTM sites, and included 30 MoRFs. In a GLTSCR1L/BICRAL-centered PPI network generated using STRING with a medium confidence level of 0.4, there were eight nodes and 13 edges (instead of seven expected edges). This network was characterized by an average node degree of 3.25 and an average local clustering coefficient of 0.911. It contained significantly more interactions than expected for the network of its size (PPI enrichment *p-*value was 0.0288).

Non-synonymous single-nucleotide variations (nsSNVs) in the *BICRAL* gene are linked to blastoma, malignant glioma, and melanoma, as well as to stomach, uterine, pancreatic, colorectal, liver, and lung cancers, whereas significant down-regulation of the *BICRAL* gene was observed in urinary bladder cancer, head and neck cancer, lung squamous cell carcinoma, breast cancer, thyroid cancer, and esophageal cancer.

## 3. Experimental Section

### 3.1. Datasets

We analyzed here the 30 major components of human BAF complex. The list of considered subunits include BRG1/SMARCA4, BRM/SMARCA2 BCL7A, BCL7B, BCL7C, BCL11A, BCL11B, BAF250A/ARID1A, BAF250B/ARID1B, BAF57/SMARCE1, BAF155/SMARCC1, BAF180/PBRM1, BAF200/ARID2, BRD7, BRD9, BAF60A/SMARCD1, BAF60B/SMARCD2, BAF60C/SMARCD3, BAF45A/PHF10, BAF45D/DPF2, BAF45B/DPF1, BAF45C/DPF3, BAF47/SMARCB1, SSXT/SS18, BAF170/SMARCC2, BAF53A/ACTL6A, BAF53B/ACTL6B, actin-β/ACTB, GLTSCR1/BICRA, and GLTSCR1L/BICRAL. Sequences of these proteins in the FASTA format and some functional information was extracted from UniProt [120]. Amino acid sequences of the analyzed proteins are shown in Appendix A.

### 3.2. Computational Characterization of Intrinsic Disorder in the BAF Complex

Initially, the subunits of human BAF complex were analyzed by five per-residue predictors, such as PONDR^®^ VLXT [61], PONDR^®^ VSL2 [259], and PONDR^®^ VL3 [259] available on the PONDR site (http://www.pondr.com), and the IUPred computational platform that allows identification of either short or long regions of intrinsic disorder, IUPred-L and IUPred-S [260]. The outputs of the evaluation of the per-residue disorder propensity by these tools are represented as real numbers between 1 (ideal prediction of disorder) and 0 (ideal prediction of order). A threshold of ≥0.5 was used to identify disordered residues and regions in query proteins. For each query protein in this study, the predicted percentage of intrinsic disorder (PPID) was calculated based on the outputs of PONDR^®^ VSL2. Here, PPID in a query protein represents a percent of residues with disorder scores exceeding 0.5.

Next, the outputs of two binary predictors, the charge-hydropathy (CH) plot [69,112] and the cumulative distribution function (CDF) plot [110,112,261] were combined to conduct a CH-CDF analysis [109,110,111,262] that allows classification of proteins based on their position within the CH-CDF phase space as ordered (proteins predicted to be ordered by both binary predictors), putative native “molten globules” or hybrid proteins (proteins determined to be ordered/compact by CH, but disordered by CDF), putative native coils and native pre-molten globules (proteins predicted to be disordered by both methods), and proteins predicted to be disordered by CH-plot, but ordered by CDF.

Complementary disorder evaluations together with important disorder-related functional information were retrieved from the D^2^P^2^ database (http://d2p2.pro/) [263], which is a database of predicted disorder for a large library of proteins from completely sequenced genomes [263]. The D^2^P^2^ database uses outputs of IUPred [260], PONDR^®^ VLXT [61], PrDOS [264], PONDR^®^ VSL2B [265,266], PV2 [263], and ESpritz [267]. The database is further supplemented by data concerning location of various curated posttranslational modifications and predicted disorder-based protein binding sites, known as molecular recognition features, MoRFs.

### 3.3. Computational Evaluation of Interactability of the Human BAF Complex Subunits

Information on the interactability of the subunits of human BAF complex was retrieved using the search tool for the retrieval of interacting genes; STRING, http://string-db.org/. STRING generates a network of protein–protein interactions based on predicted and experimentally validated information on the interaction partners of a protein of interest [268]. In the corresponding network, the nodes correspond to proteins, whereas the edges show predicted or known functional associations. Seven types of evidence are used to build the corresponding network, where they are indicated by the differently colored lines: A green line represents neighborhood evidence; a red line—the presence of fusion evidence; a purple line—experimental evidence; a blue line—co-occurrence evidence; a light blue line—database evidence; a yellow line—text mining evidence; and a black line—co-expression evidence [268]. In this study, STRING was utilized in three different modes—to generate the network of the inter-BAF-complex PPI interactions, to produce the BAF-centered PPI network, and to create PPI networks centered at individual subunits of the human BAF complex. These individual networks for each subunit of human BAF complex can be found in the Appendix A. Resulting PPI networks were further analyzed using STRING-embedded routines in order to retrieve the network-related statistics, such as: The number of nodes (proteins); the number of edges (interactions); average node degree (average number of interactions per protein); average local clustering coefficient, which defines how close the neighbors are to being a complete clique: If a local clustering coefficient is equal to 1, then every neighbor connected to a given node *N_i_* is also connected to every other node within the neighborhood, and if it is equal to 0, then no node that is connected to a given node *N_i_* connects to any other node that is connected to *N_i_*; expected number of edges (which is a number of interactions among the proteins in a random set of proteins of similar size); and a PPI enrichment *p*-value (which is a reflection of the fact that query proteins in the analyzed PPI network have more interactions among themselves than what would be expected for a random set of proteins of similar size, drawn from the genome. It was pointed out that such an enrichment indicates that the proteins are at least partially biologically connected, as a group).

## 4. Conclusions

This article presents the results of unprecedented computational and bioinformatics analysis of the intrinsic disorder predisposition of the subunits of chromatin remodeling complex BAF. We show here that as a rule, the constituents of this proteinaceous machine contain very high levels of intrinsic disorder. As a matter of fact, the vast majority of human BAF subunits are highly disordered proteins characterized by PPID values exceeding 50%. These observations define a set of BAF subunits as one the most disordered set of functionally related proteins. With no exception, all BAF subunits contain functionally important IDPRs. Disorder in these proteins is used for protein–protein interactions and also serves as a signal for a variety of posttranslational modifications. This information adds a new angle of complexity to this machine, which is known to be characterized by a highly polymorphic structure, containing from four to 17 subunits encoded by 29 genes. Our data indicated that this compositional polymorphism of BAF complex was further intensified by the structural polymorphism of its subunits defined by their high disorder contents, the presence of multiple PTM sites, and numerous isoforms generated by alternative splicing. These features (intrinsic disorder, PTMs, and alternative splicing) were considered as means for the generation of various proteoforms (i.e., a set of structurally and functionally distinct protein molecules encoded by a single gene) [269,270], which constitute a foundation for the “protein structure-function continuum” model, where a given protein exists as a dynamic conformational ensemble containing multiple proteoforms (conformational/basic, inducible/modified, and functioning) characterized by a broad spectrum of structural features and possessing various functional potentials [269,271,272]. Therefore, it seems that the BAF complex represents the realization of the proteoform concept at the proteinaceous machine level. This disorder-based structural and functional heterogeneity of BAF subunits also provides an explanation for the common involvement of these proteins in various human diseases. In fact, this is in line with the common knowledge that many disease-associated proteins are IDPs or hybrid proteins containing ordered domains and functional IDPRs [273,274]: Since IDPs/IDPRs are commonly involved in recognition, regulation, and cell signaling, their deregulation and misfolding can lead to misidentification, misinteractions, and missignaling and, therefore, can be pathogenic [273,275,276,277,278,279]. In other words, the data presented in this study on the prevalence of intrinsic disorder in the subunits of human chromatin remodeling complex BAF opened another level of sophistication in the possible routes of BAF pathogenicity.

## Figures and Tables

**Figure 1 ijms-20-05260-f001:**
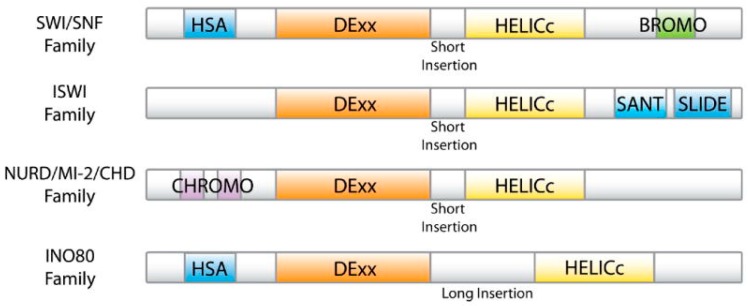
Domain organization of the ATPase subunits found in different ATP-dependent chromatin remodeling complexes [12].

**Figure 2 ijms-20-05260-f002:**
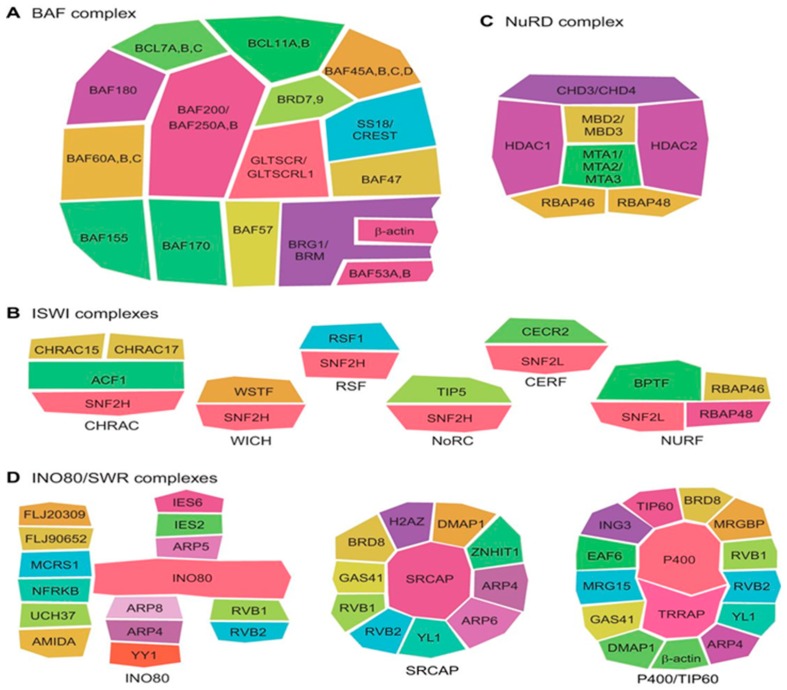
Subunit organization of different ATP-dependent chromatin remodeling complexes. Reproduced/adapted with permission Hota et al. 2019 [14]. Dynamic BAF chromatin remodeling complex subunit inclusion promotes temporally distinct gene expression programs in cardiogenesis [14]. Compositions of BAF complex (**A**), various ISWI complexes (**B**), NuRD complex (**C**), and different INO80/SWR complexes (**D**) are shown.

**Figure 3 ijms-20-05260-f003:**
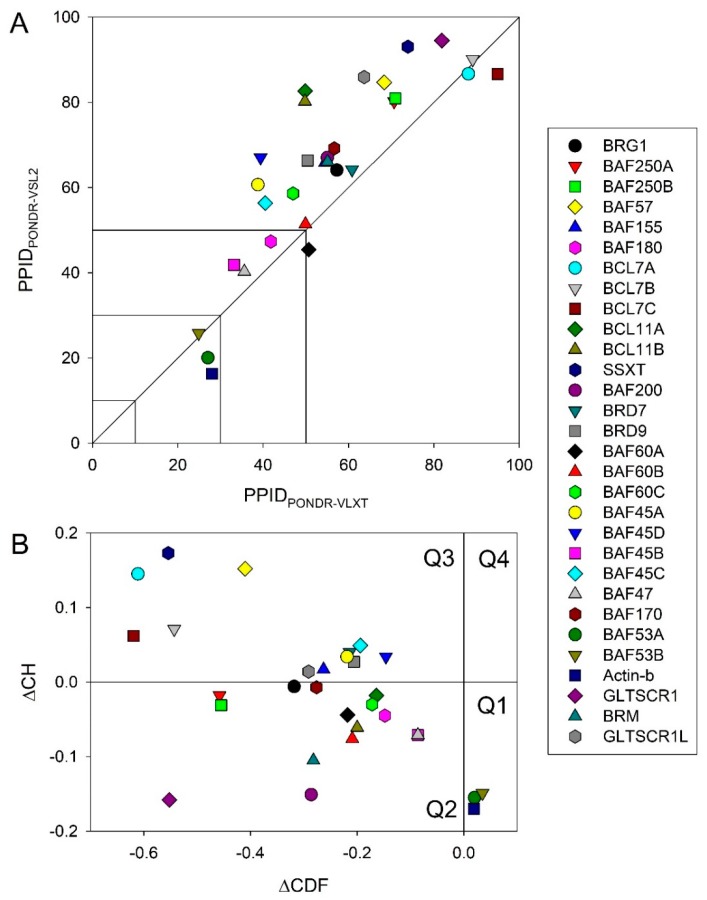
Evaluation of the overall disorder status of 30 subunits of the human BAF complex. (**A**) 2D disorder plot showing presenting the PPID_PONDR_^®^_VSL2_ vs. PPID _PONDR_^®^_VLXT_ dependence. (**B**) Charge-hydropathy cumulative distribution function (CH-CDF) plot. In this plot, the coordinates of each spot are calculated as a distance of the corresponding protein in the CH-plot from the boundary (Y-coordinate) and an average distance of the respective CDF curve from the CDF boundary (X-coordinate) [109,110,111].

**Figure 4 ijms-20-05260-f004:**
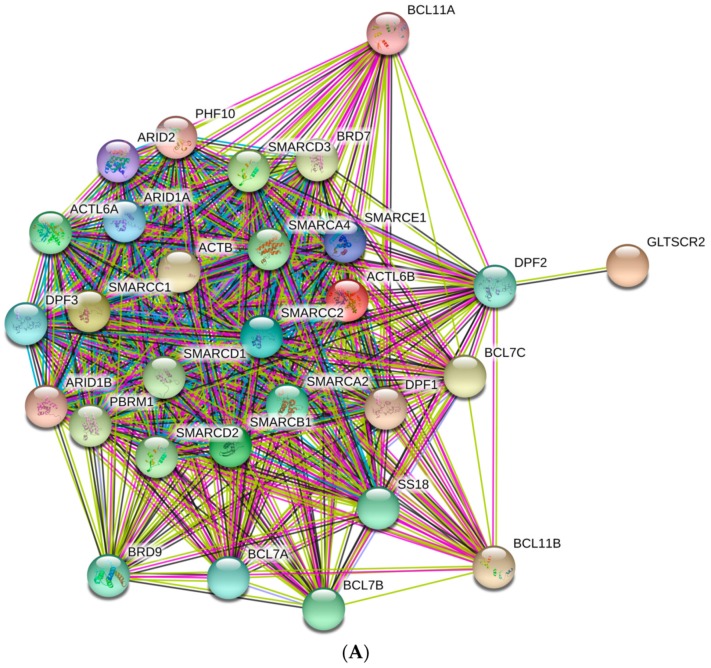
Evaluation of the interactability of human BAF subunits by the STRING platform. (**A)** Network of the inter-complex protein–protein interactions (PPIs; 30 subunits, low confidence level of 0.15). (**B**) BAF-centered PPI network (530 proteins, high confidence level of 0.7).

**Figure 5 ijms-20-05260-f005:**
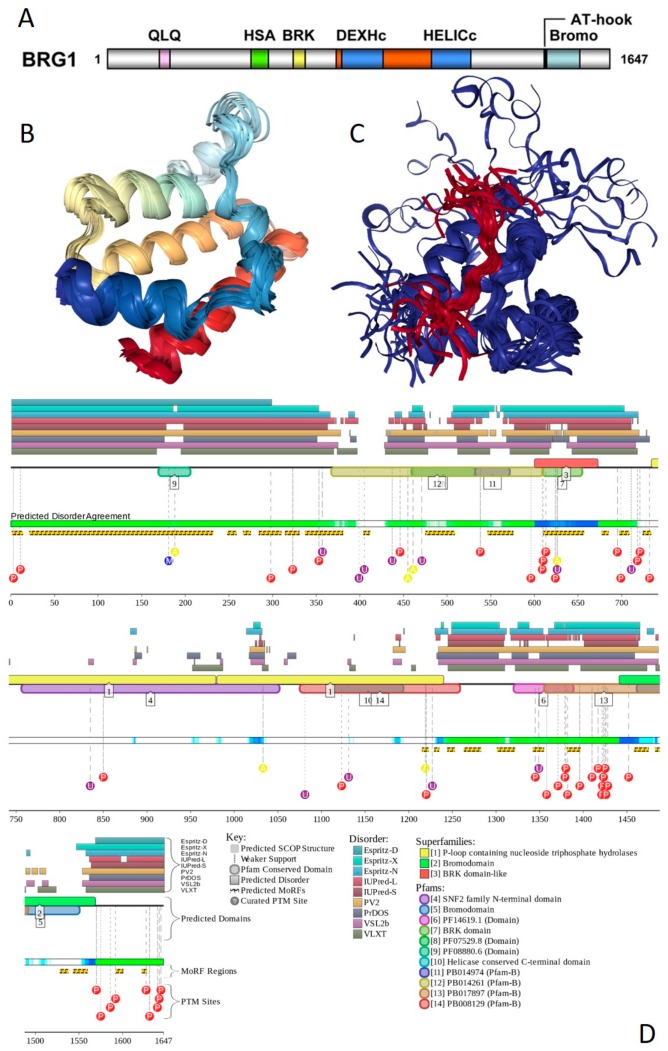
Structural organization of human BRG1 protein. (**A**) Domain architecture of the protein. (**B**) NMR solution structure of the bromodomain (residues 1452–1570, PDB ID: 2H60). (**C**) NMR solution structure of the C-terminal peptide of BRG1 (residues 1591–1602, red structures) bound to the Brd3 ET domain (PDB ID: 6BGH, blue structures). (**D**) D^2^P^2^ functional disorder profile of human BRG1 protein (UniProt ID: P51532).

**Figure 6 ijms-20-05260-f006:**
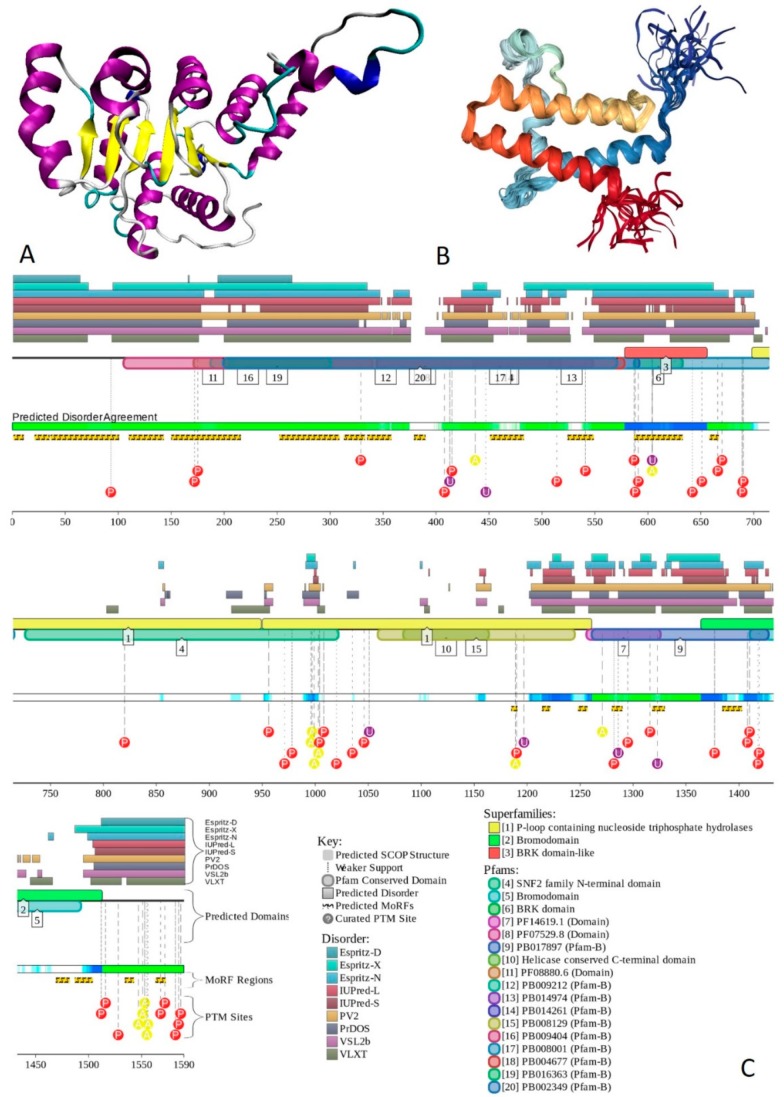
Structural organization of human BRM protein. (**A**) X-ray crystal structure of helicase ATP-binding domain (residues 705–955; PDB ID: 6EG3). (**B**) NMR solution structure of the bromodomain (residues 1477–1504, PDB ID: 2DAT). **C**. D^2^P^2^ functional disorder profile of the human BRM protein (UniProt ID: P51531).

**Figure 7 ijms-20-05260-f007:**
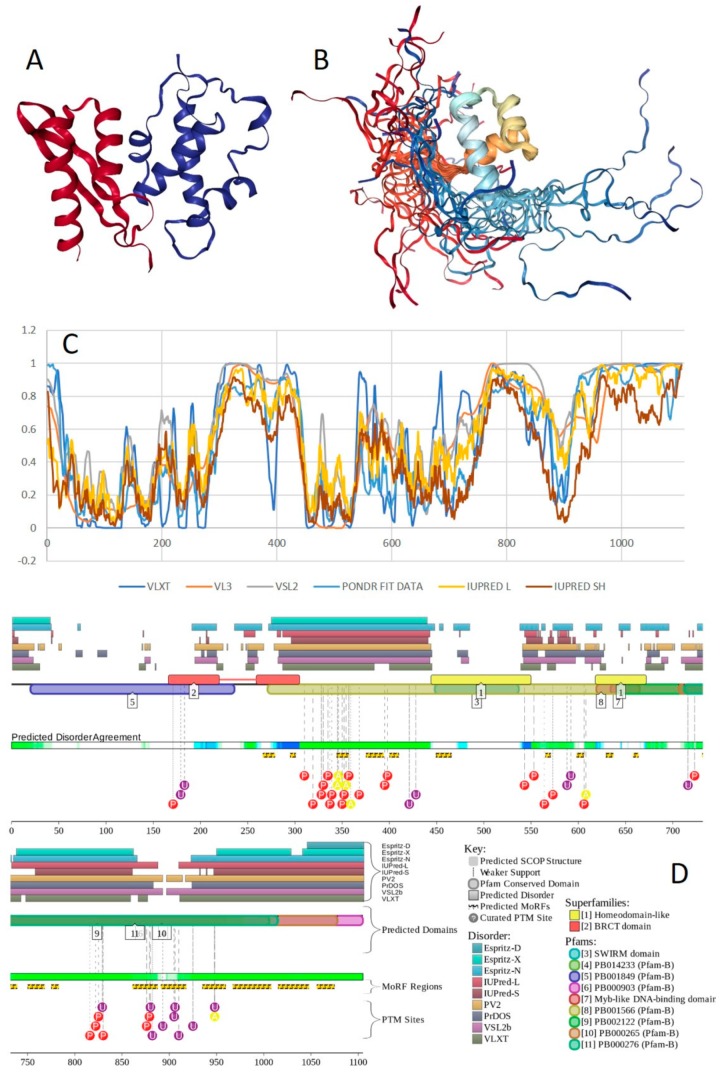
Structural organization of the human BAF155 protein. (**A**) X-ray crystal structure of SWIRM domain (residues 446–540, blue structure) in a complex with the repeat 1 (RPT1) domain of BAF47 (residues 183–289, red structure; PDB ID: 5GJK). (**B**) NMR solution structure of the SANT domain (residues 610–675; PDB ID: 1RYU). (**C**) Intrinsic disorder profile of human BAF155 generated by six commonly used predictors of intrinsic disorder (PONDR^®^ VLXT, PONDR^®^ VL3, PONDR^®^ VSL2, PONDR^®^ FIT, IUPred_short, and IUPred_long). (**D**) D^2^P^2^ functional disorder profile of the human BAF155 protein (UniProt ID: Q92922).

**Figure 8 ijms-20-05260-f008:**
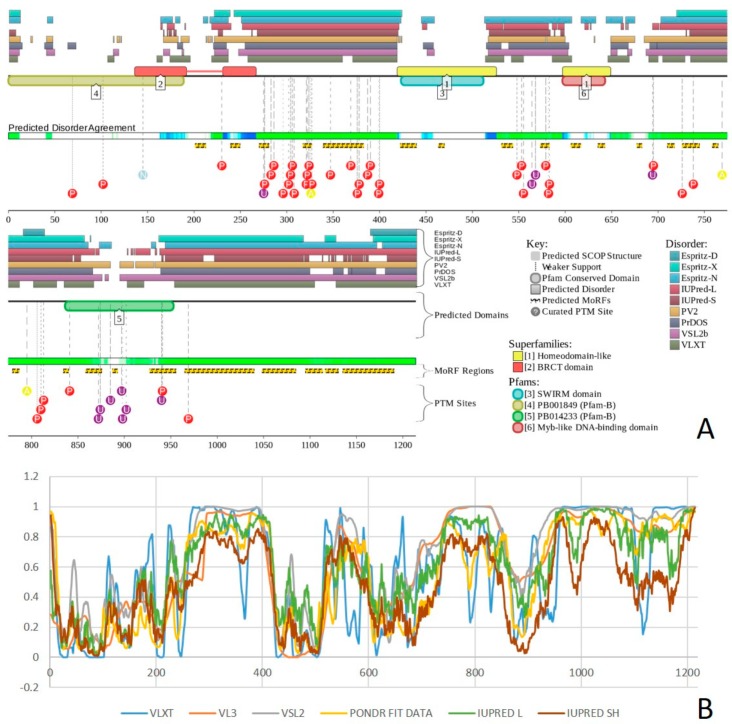
Peculiarities of disorder distribution in the human BAF170 protein. (**A**) D^2^P^2^ functional disorder profile of human BAF170 protein (UniProt ID: Q8TAQ2). (**B**) Intrinsic disorder profile of human BAF170 generated by six commonly used predictors of intrinsic disorder (PONDR^®^ VLXT, PONDR^®^ VL3, PONDR^®^ VSL2, PONDR^®^ FIT, IUPred_short, and IUPred_long).

**Figure 9 ijms-20-05260-f009:**
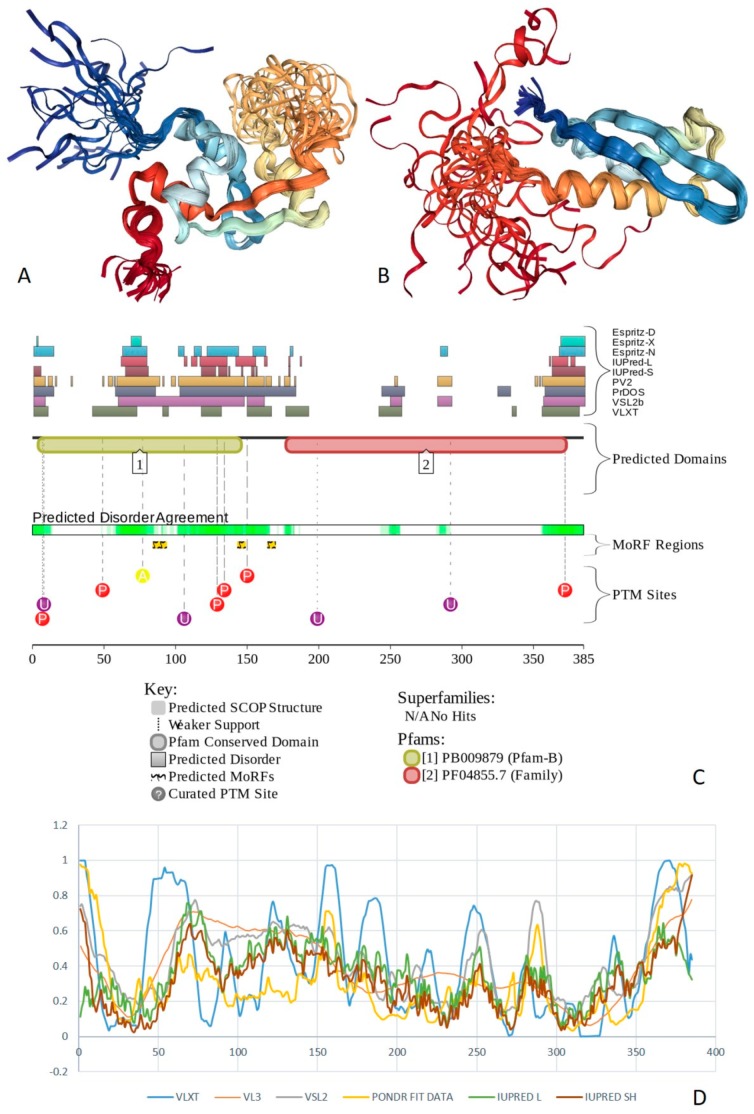
Structural organization of the human BAF47 protein. (**A**) NMR structure of the DNA-binding domain (residues 2–113, PDB ID: 5AJ1). (**B**) NMR solution structure of the MYC/HIV-1 integrase binding region (residues 183–265; PDB ID: 6AX5). (**C**) D^2^P^2^ functional disorder profile of human BAF47 protein (UniProt ID: Q12824). (**D**) Intrinsic disorder profile of human BAF47 generated by six commonly used predictors of intrinsic disorder (PONDR^®^ VLXT, PONDR^®^ VL3, PONDR^®^ VSL2, PONDR^®^ FIT, IUPred_short, and IUPred_long).

**Figure 10 ijms-20-05260-f010:**
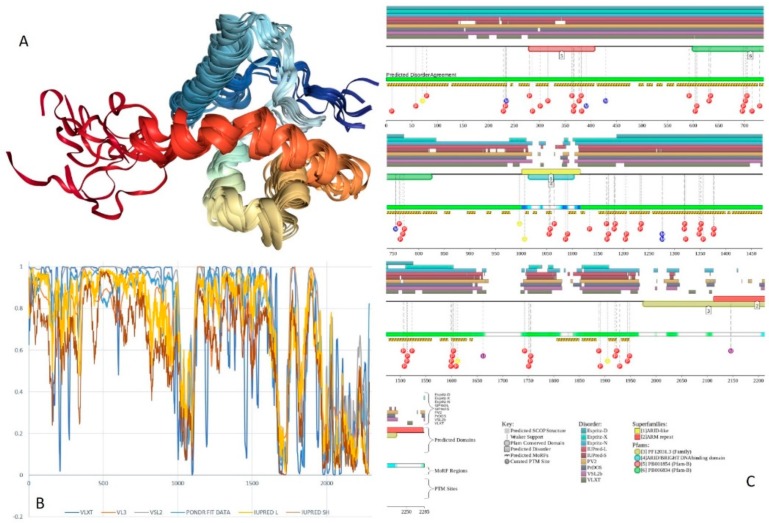
Structural organization of the human BAF250A protein. (**A**) NMR solution structure of the ARID domain (residues 1017–1108; PDB ID: 1RYU). (**B**) Intrinsic disorder profile of human BAF250A generated by six commonly used predictors of intrinsic disorder (PONDR^®^ VLXT, PONDR^®^ VL3, PONDR^®^ VSL2, PONDR^®^ FIT, IUPred_short, and IUPred_long) (**C**) D^2^P^2^ functional disorder profile of the human BAF250A protein (UniProt ID: O14497).

**Figure 11 ijms-20-05260-f011:**
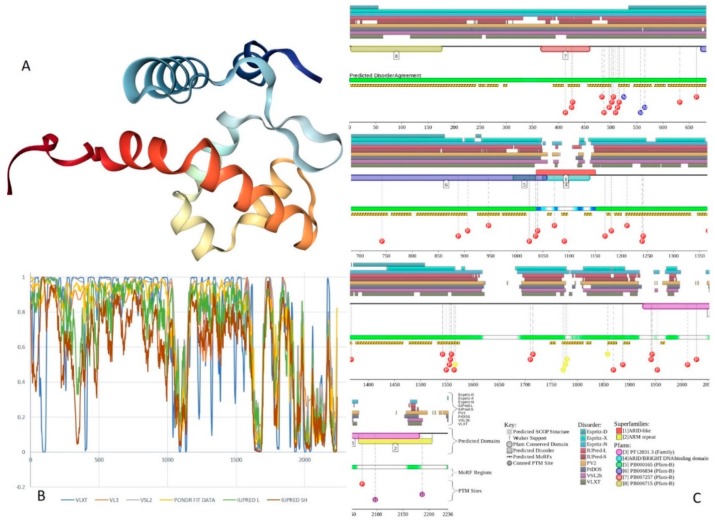
Structural organization of the human BAF250B protein. (**A**) X-ray crystal structure of the ARID domain (residues 1041–1159; PDB ID: 2CXY). (**B**) Intrinsic disorder profile of human BAF250B protein (UniProt ID: Q8NFD5) generated by six commonly used predictors of intrinsic disorder (PONDR^®^ VLXT, PONDR^®^ VL3, PONDR^®^ VSL2, PONDR^®^ FIT, IUPred_short, and IUPred_long). (**C**) D^2^P^2^ functional disorder profile of the human BAF250B protein.

**Figure 12 ijms-20-05260-f012:**
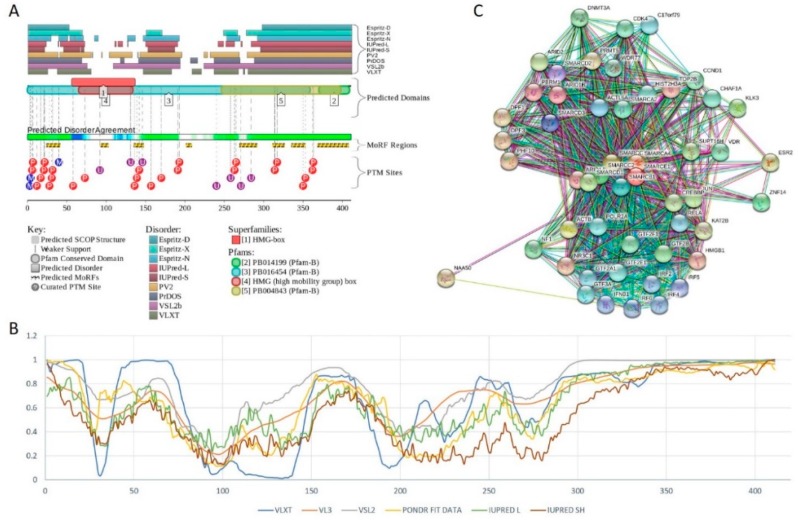
Structural and functional analysis of the human BAF57 protein (UniProt ID: Q969G3). (**A**) D^2^P^2^ functional disorder profile of human BAF57 protein. (**B**) Intrinsic disorder profile of human BAF57 protein generated by six commonly used predictors of intrinsic disorder (PONDR^®^ VLXT, PONDR^®^ VL3, PONDR^®^ VSL2, PONDR^®^ FIT, IUPred_short, and IUPred_long). (**C**) STRING-generated protein–protein interaction network of human BAF57.

**Figure 13 ijms-20-05260-f013:**
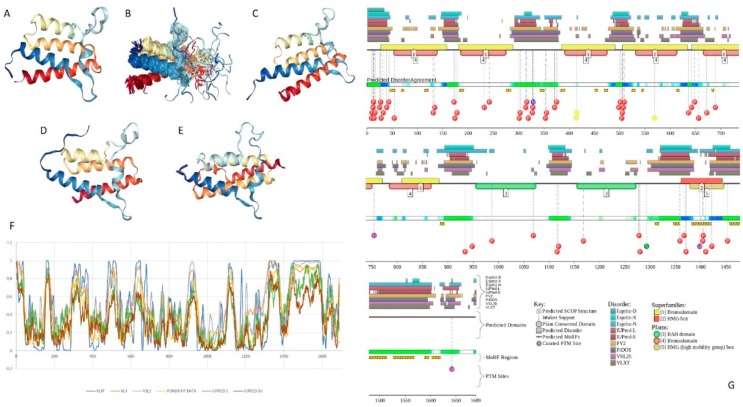
Structural organization of the human BAF180 protein. (**A**) X-ray crystal structure of the first bromodomain (residues 34–154; PDB ID: 3IU5). (**B**) NMR solution structure of the second bromodomain (residues 174–292; PDB ID: 2KTB). (**C**) X-ray crystal structure of the third bromodomain (residues 388–494; PDB ID: 3K2J). (**D**) X-ray crystal structure of the fifth bromodomain (residues 645–766; PDB ID: 3GoJ). (**E**) X-ray crystal structure of the sixth bromodomain (residues 773–914; PDB ID: 3IU6). (**F**) Intrinsic disorder profile of human BAF180 protein (UniProt ID: Q86U86) generated by six commonly used predictors of intrinsic disorder (PONDR^®^ VLXT, PONDR^®^ VL3, PONDR^®^ VSL2, PONDR^®^ FIT, IUPred_short, and IUPred_long). (**G**) D^2^P^2^ functional disorder profile of the human BAF180 protein.

**Figure 14 ijms-20-05260-f014:**
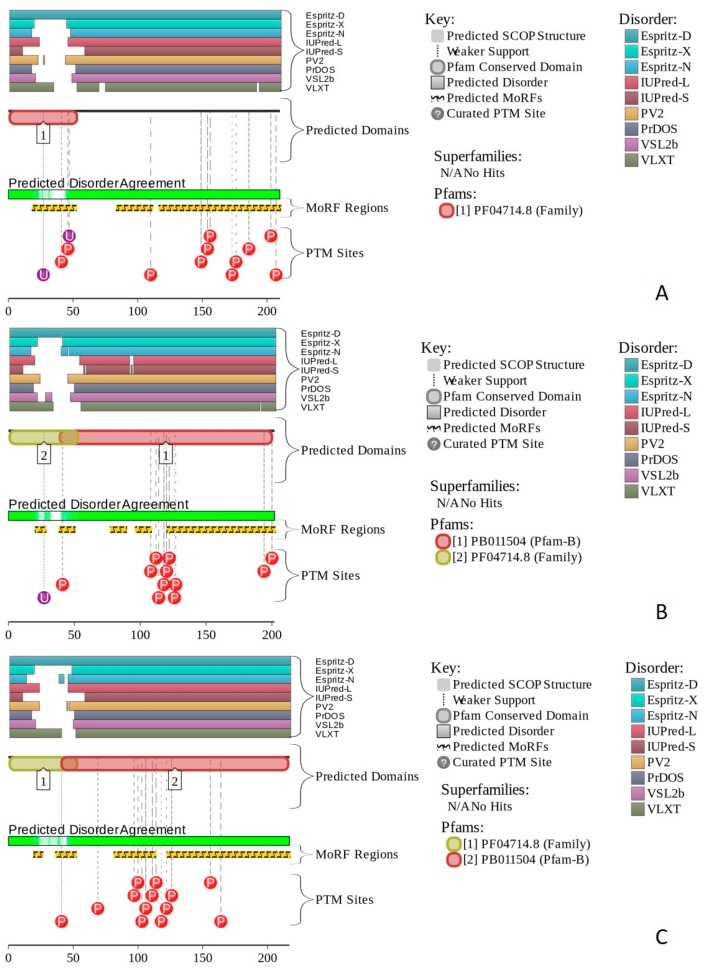
Intrinsic disorder and functionality of human BCL7A (UniProt ID: Q4VC05) (**A**), BCL7B (UniProt ID: Q9BQE9) (**B**), and BCL7C (UniProt ID: Q8WUZ0) (**C**) as evidenced by the D^2^P^2^ platform.

**Figure 15 ijms-20-05260-f015:**
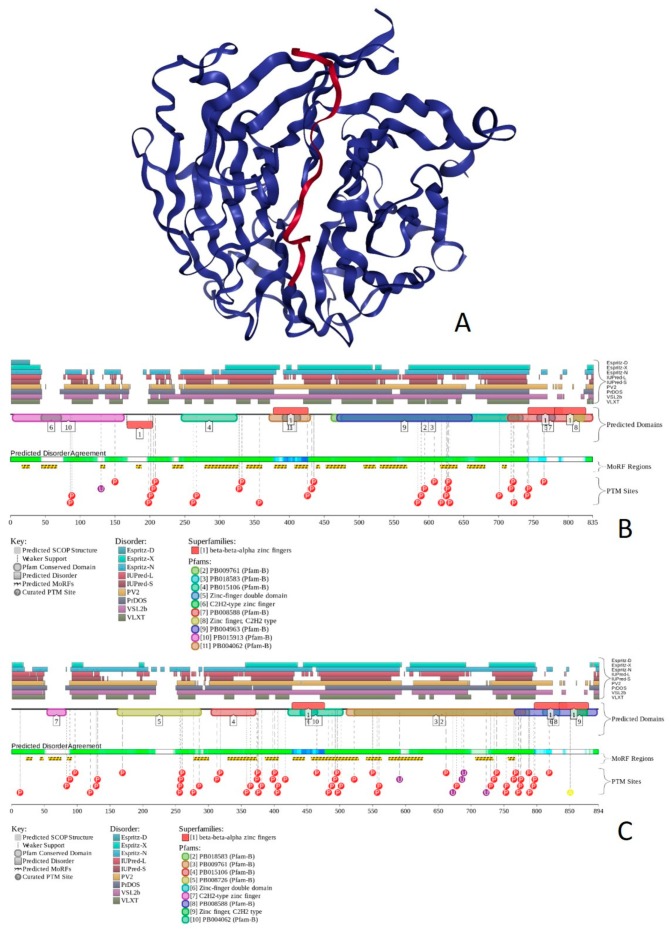
Structure, intrinsic disorder, and functionality of human BCL11A and BCL11B. (**A**) X-ray crystal structure of the N-terminal peptide of BCL11A (red structure) complexed with the histone-binding protein RBBP4 (blue structure; PDB ID: 5VTB). (**B**) D^2^P^2^ functional disorder profile of human BCL11A (UniProt ID: Q9H165). (**C**) D^2^P^2^ functional disorder profile of human BCL11BA (UniProt ID: Q9C0K0).

**Figure 16 ijms-20-05260-f016:**
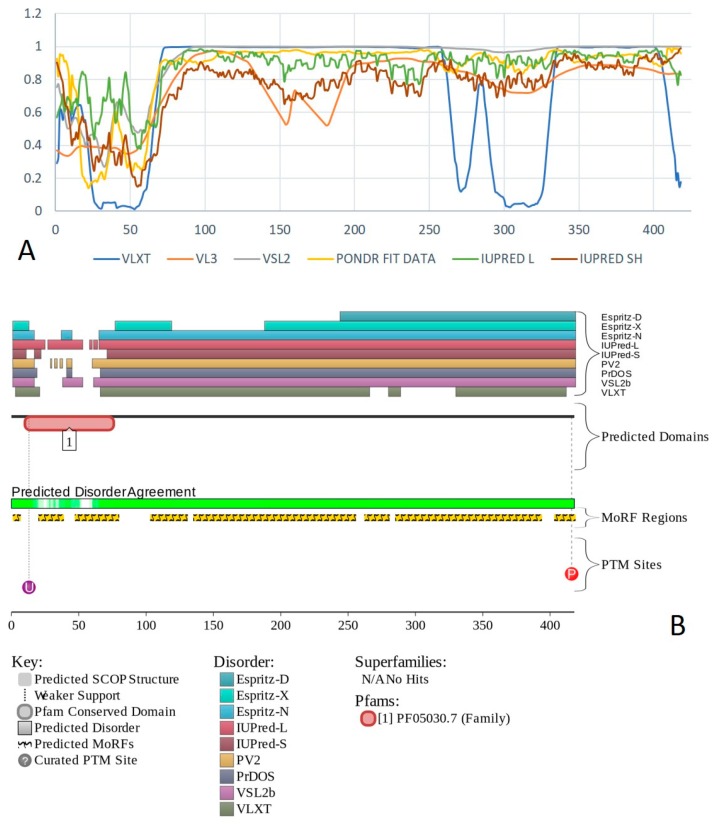
Structural and functional analysis of the human SSXT protein (UniProt ID: Q15532). (**A**) Intrinsic disorder profile of human SSXT protein generated by six commonly used predictors of intrinsic disorder (PONDR^®^ VLXT, PONDR^®^ VL3, PONDR^®^ VSL2, PONDR^®^ FIT, IUPred_short, and IUPred_long). (**B**) D^2^P^2^ functional disorder profile of the human SSXT protein.

**Figure 17 ijms-20-05260-f017:**
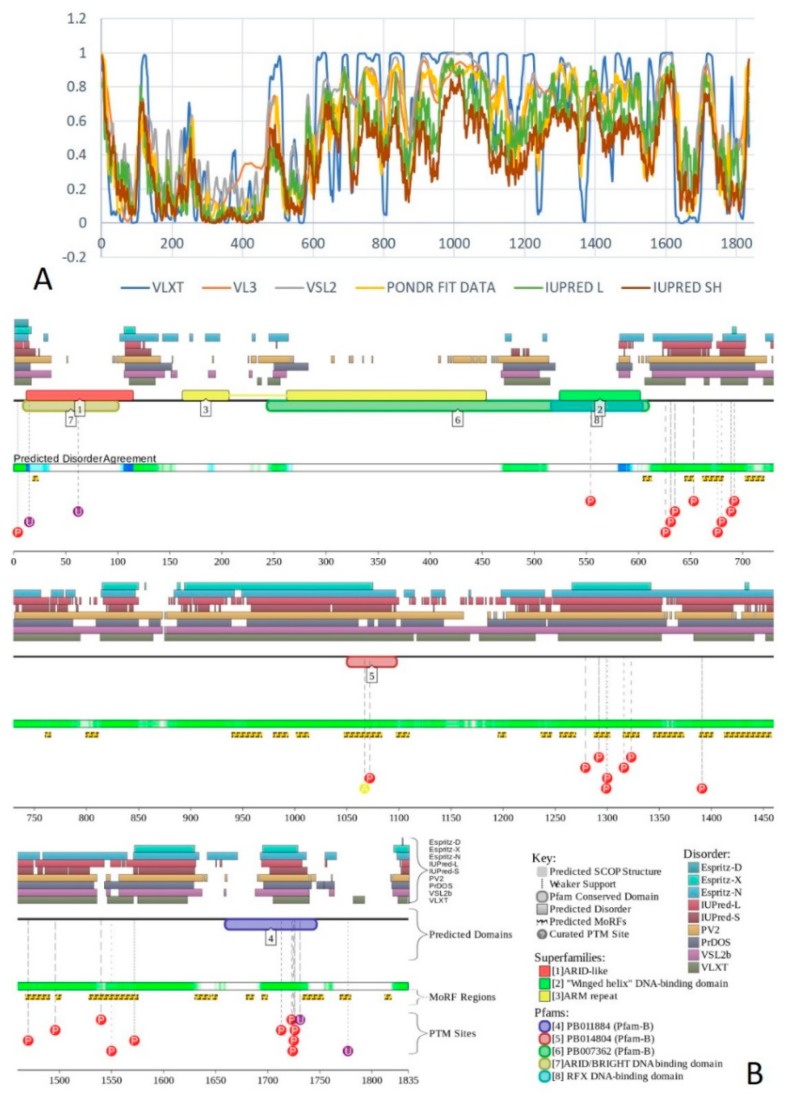
Structural and functional analysis of the human BAF200/ARID2 protein (UniProt ID: Q68CP9). (**A**) Intrinsic disorder profile of human BAF200/ARID2 protein generated by six commonly used predictors of intrinsic disorder (PONDR^®^ VLXT, PONDR^®^ VL3, PONDR^®^ VSL2, PONDR^®^ FIT, IUPred_short, and IUPred_long). (**B**) D^2^P^2^ functional disorder profile of the human BAF200/ARID2 protein.

**Figure 18 ijms-20-05260-f018:**
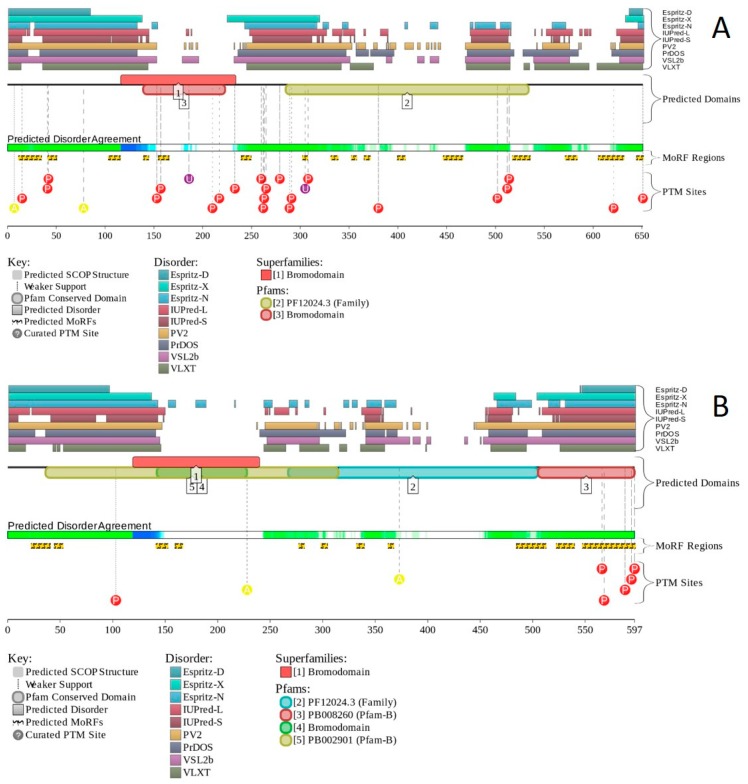
Structure, intrinsic disorder, and functionality of human BRD7 and BRD9. (**A**) D^2^P^2^ functional disorder profile of human BRD7 (UniProt ID: Q9NPI1). (**B**) D^2^P^2^ functional disorder profile of human BRD9 (UniProt ID: Q9H8M2).

**Figure 19 ijms-20-05260-f019:**
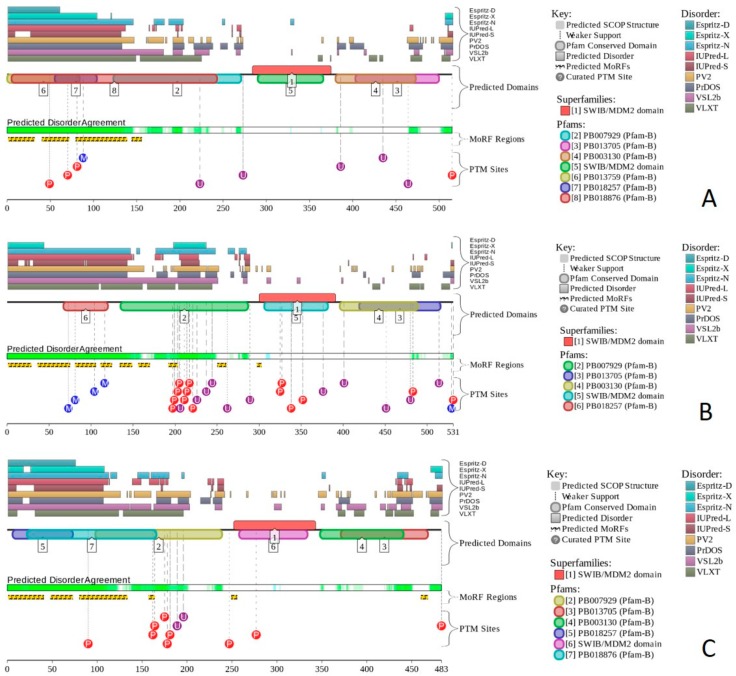
Structure, intrinsic disorder, and functionality of human BAF60A, BAF60B, and BAF60C proteins. (**A**) D^2^P^2^ functional disorder profile of human BAF60A (UniProt ID: Q96GM5). (**B**) D^2^P^2^ functional disorder profile of human BAF60B (UniProt ID: Q92925). (**C**) D^2^P^2^ functional disorder profile of human BAF60C (UniProt ID: Q6STE5).

**Figure 20 ijms-20-05260-f020:**
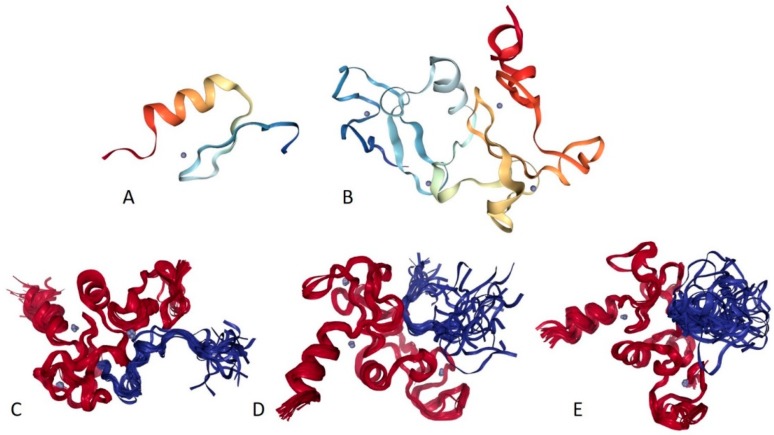
Structural characterization of human DPF2/BAF45D and DPF3/BAF45C proteins. X-ray crystal structures of C2H2-type zinc finger domain (**A**; PDB ID: 3IUF) and double PHD finger domain of human DPF2/BAF45D (**B**; PDB ID: 5B79). NMR solution structures of the double PHD fingers of human transcriptional protein DPF3/BAF45C (red structures residues 261–372) bound to: (**C**) Histone H3 peptide acetylated at lysine 14 (PDB ID: 2KWJ; blue structures); (**D**) Histone H4 peptide containing acetylation at lysine 16 (PDB ID: 2KWN; blue structures); and (**E**) Histone H4 peptide containing N-terminal acetylation at serine 1 (PDB ID: 2KWO; blue structures).

**Figure 21 ijms-20-05260-f021:**
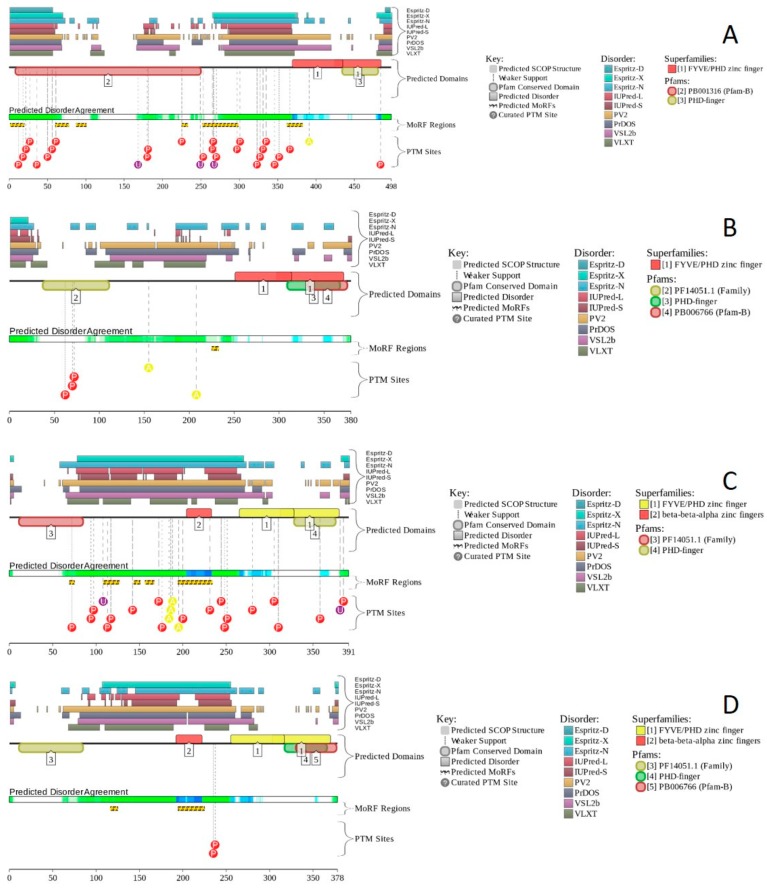
Structure, intrinsic disorder, and functionality of human BAF45A, BAF45B, BAF45C, and BAF45D proteins. (**A**) D^2^P^2^ functional disorder profile of human PHF10/BAF45A (UniProt ID: Q8WUB8). (**B**) D^2^P^2^ functional disorder profile of human DPF1/BAF45B (UniProt ID: Q92782). (**C**) D^2^P^2^ functional disorder profile of human DPF2/BAF45D (UniProt ID: Q92785). (**D**) D^2^P^2^ functional disorder profile of human DPF3/BAF45C (UniProt ID: Q92784).

**Figure 22 ijms-20-05260-f022:**
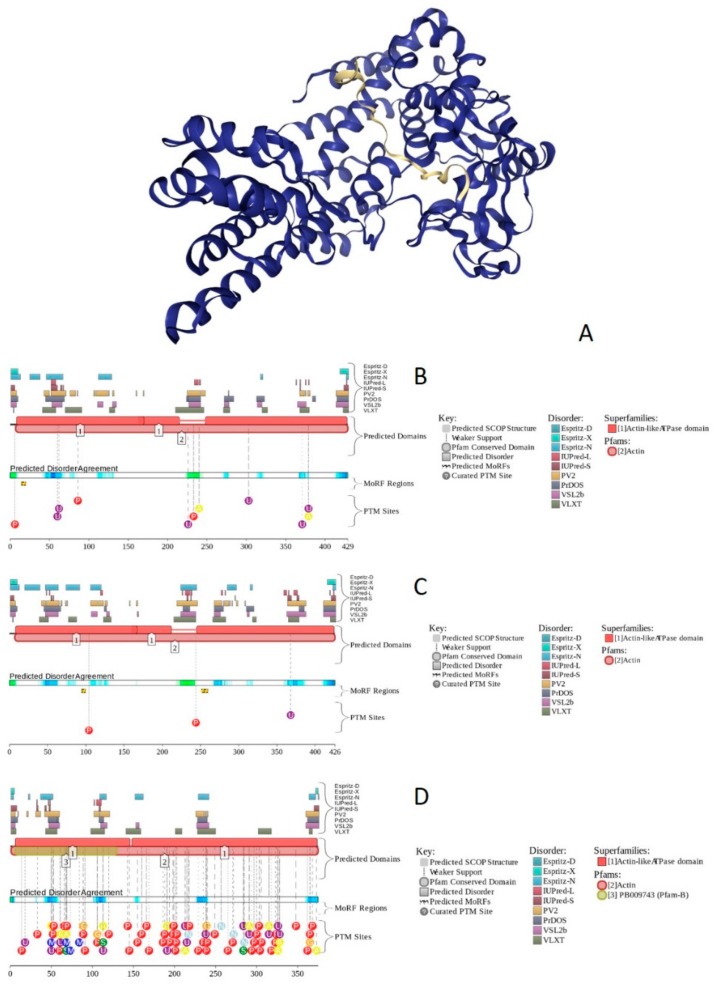
Structure, intrinsic disorder, and functionality of human BAF53A/ACTL6A, BAF53B/ACTL6B, and β-actin/ACTB. (**A**) X-ray crystal structure of a short actin peptide (residues 66–88, yellow structure) co-crystallized with human histone-lysine N-methyltransferase (blue structure) (PDB ID: 6ICT). (**B)** D^2^P^2^ functional disorder profile of human BAF53A/ACTL6A (UniProt ID: O96019). (**C**) D^2^P^2^ functional disorder profile of human BAF53B/ACTL6B (UniProt ID: O94805). (**D**) D^2^P^2^ functional disorder profile of human β-actin/ACTB (UniProt ID: P60709).

**Figure 23 ijms-20-05260-f023:**
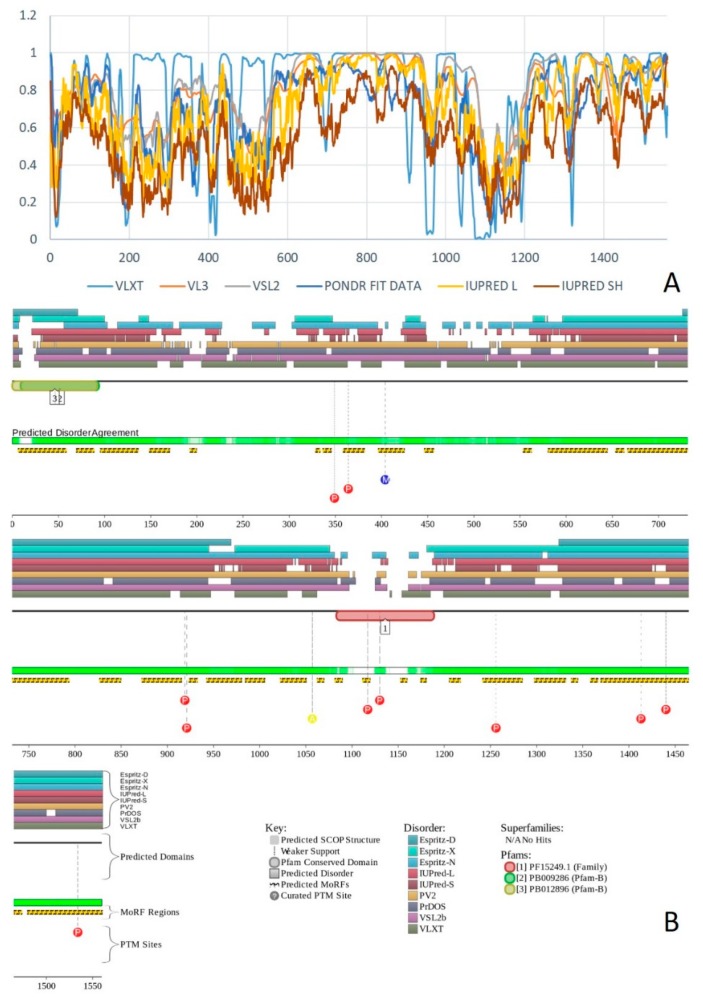
Structural and functional analysis of the human GLTSCR1/BICRA protein (UniProt ID: Q9NZM4). (**A**) Intrinsic disorder profile of human GLTSCR1/BICRA protein generated by six commonly used predictors of intrinsic disorder (PONDR^®^ VLXT, PONDR^®^ VL3, PONDR^®^ VSL2, PONDR^®^ FIT, IUPred_short, and IUPred_long). (**B**) D^2^P^2^ functional disorder profile of the human GLTSCR1/BICRA protein.

**Figure 24 ijms-20-05260-f024:**
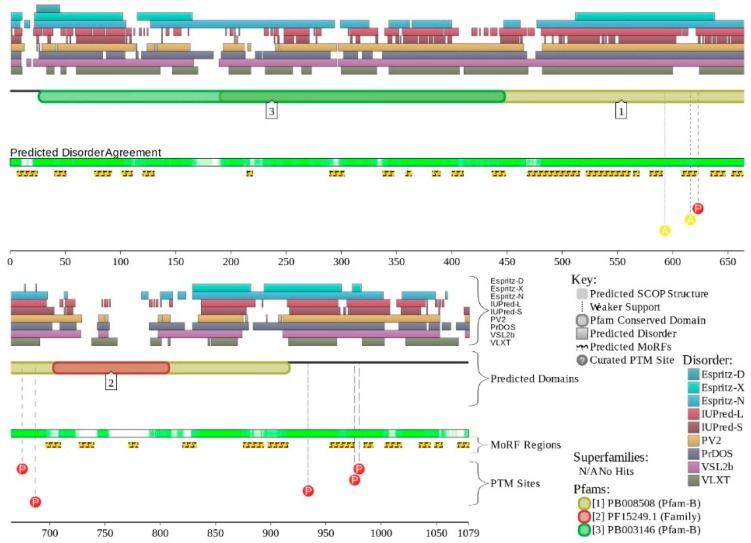
D^2^P^2^ functional disorder profile of human GLTSCR1L/BICRAL protein (UniProt ID: Q6AI39).

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
