# Peer review of "Intrinsic Disorder of the BAF Complex: Roles in Chromatin Remodeling and Disease Development"

_ijms, 2019, doi:10.3390/ijms20215260_

Round 1

Reviewer 1 Report

General comments:

Authors should revise the whole manuscript to improve its readability. In addition to grammatical and typographical errors, there are several long and complicated sentences.

For example, “Among such ATP-dependent nucleosome remodelers, which are considered as important players of transcriptional regulation of many eukaryotic genes, are polymorphic BRG-/BRM-Associated Factor (BAF) and Polybromo-associated BAF (PBAF) complexes are the members of the SWItch/Sucrose Non-Fermentable (SWI/SNF) family of the ATP-dependent chromatin-remodeling complexes that are essential for mammalian transcription and development”.

Similarly, presentation of figures should also be improved and simplified, specifically the D2P2 profiles. There is a level of redundancy (e.g. outputs from Disorder prediction program) and relative to the information content in figure, the discussion in the text is much limited.

For a few of the BAF subunits, authors have pointed out compositional biases in the amino acid sequence e.g. Pro-rich and Glu-rich regions. It was not clear if other subunits, discussed in the manuscript, have similar compositional biases and also how these compositional biases influence the protein-protein interaction. 

Author Response

Authors should revise the whole manuscript to improve its readability. In addition to grammatical and typographical errors, there are several long and complicated sentences.

For example, “Among such ATP-dependent nucleosome remodelers, which are considered as important players of transcriptional regulation of many eukaryotic genes, are polymorphic BRG-/BRM-Associated Factor (BAF) and Polybromo-associated BAF (PBAF) complexes are the members of the SWItch/Sucrose Non-Fermentable (SWI/SNF) family of the ATP-dependent chromatin-remodeling complexes that are essential for mammalian transcription and development”.

REPLY: Thank you for pointing this out. The manuscript was carefully edited by a native English speaker. All lines, in which corrections were made, are highlighted in yellow.

Similarly, presentation of figures should also be improved and simplified, specifically the D2P2 profiles. There is a level of redundancy (e.g. outputs from Disorder prediction program) and relative to the information content in figure, the discussion in the text is much limited.

REPLY: We respectfully disagree with this request. In our view, presented data are important for the manuscript. Although the reviewer states that there is redundancy in the outputs from disorder prediction programs, this redundancy is apparent, since protein intrinsic disorder is a very heterogeneous phenomenon and different disorder predictors are focused in different aspects of this phenomenon. The outputs of D2P2 (the D2P2 profiles) cannot be changed, since they are generated by the D2P2 platform which uses pre-default settings that cannot be adjusted.

For a few of the BAF subunits, authors have pointed out compositional biases in the amino acid sequence e.g. Pro-rich and Glu-rich regions. It was not clear if other subunits, discussed in the manuscript, have similar compositional biases and also how these compositional biases influence the protein-protein interaction. 

REPLY: Thank you for pointing this out. We carefully checked all BAF subunits discussed in this article for the presence of regions with compositional biases in the amino acid sequence. The corresponding information is added to the revised manuscript. All BAF subunits, for which compositional biases in the amino acid sequence were reported, are discussed in the manuscript now. The corresponding added information is highlighted in green. In several cases, such regions overlap or are included into the MoRFs, indicating that compositional biases can be related to protein interactability. 

Reviewer 2 Report

The authors report on the role of the BAF complex in chromatin remodeling by using computational and bioinformatics tools. Their CH-CDF analysis suggests that at least 27 out of the 30 analyzed constituents of the BAF complex are highly disordered. For each of the 30 analyzed proteins, they also give a nice overview on the structural and biological data reported in the literature (NMR or crystal structures, PTMs, mutations, expression profile and impact in cancer), and additionally show the functional disorder profile.

The manuscript is very well written and provides a very accurate analysis of 30 of the constituents of the BAF complex. Its publication is recommended as it is.

Author Response

The authors report on the role of the BAF complex in chromatin remodeling by using computational and bioinformatics tools. Their CH-CDF analysis suggests that at least 27 out of the 30 analyzed constituents of the BAF complex are highly disordered. For each of the 30 analyzed proteins, they also give a nice overview on the structural and biological data reported in the literature (NMR or crystal structures, PTMs, mutations, expression profile and impact in cancer), and additionally show the functional disorder profile.

The manuscript is very well written and provides a very accurate analysis of 30 of the constituents of the BAF complex. Its publication is recommended as it is.

REPLY: We are thankful to this reviewer for careful reading of the manuscript and high evaluation of our work.

Round 2

Reviewer 1 Report

I am pleased with the efforts from Authors to improve the manuscript and recommend its publication in the present form.